# Quantifying the role of transcript levels in mediating DNA methylation effects on complex traits and diseases

Marie C. Sadler [1,2,3] ✉, Chiara Auwerx [1,2,3,4], Kaido Lepik[1,2,3], Eleonora Porcu [1,2,3,4,5] & Zoltán Kutalik [1,2,3,5] ✉

High-dimensional omics datasets provide valuable resources to determine the causal role of molecular traits in mediating the path from genotype to phenotype. Making use of molecular quantitative trait loci (QTL) and genome-wide association study (GWAS) summary statistics, we propose a multivariable Mendelian randomization (MVMR) framework to quantify the proportion of the impact of the DNA methylome (DNAm) on complex traits that is propagated through the assayed transcriptome. Evaluating 50 complex traits, we find that on average at least 28.3% (95% CI: [26.9%–29.8%]) of DNAm-to-trait effects are mediated through (typically multiple) transcripts in the *cis*-region. Several regulatory mechanisms are hypothesized, including methylation of the promoter probe cg10385390 (chr1:8'022'505) increasing the risk for inflammatory bowel disease by reducing *PARK7* expression. The proposed integrative framework can be extended to other omics layers to identify causal molecular chains, providing a powerful tool to map and interpret GWAS signals.

In the past decade, genome-wide association studies (GWASs) have identified thousands of genetic variants associated with complex traits[1], however, linking these variants to molecular pathways still remains challenging[2]. GWAS signals of common diseases predominantly fall into the non-coding genome[3] and both their enrichment in regulatory elements (e.g., quantitative trait loci (QTL)[3,4]), as well as advances in omics technology[5], have motivated the establishment of large-scale consortia providing publicly available QTL datasets for molecular phenotypes such as DNA methylation (DNAm)[6], transcript[7,8], protein[9–11] and metabolite[12,13] levels.

Integrative statistical methods combining GWAS and omics QTL summary data include colocalization tests[14,15], summary versions of transcriptome-wide association studies (TWAS)[16,17] and Mendelian randomization (MR) studies[18,19]. Colocalization methods identify shared QTL and GWAS signals, and while this might indicate causality between the molecular and GWAS trait, signal overlap can also arise

due to reverse causality (i.e., causal effect of the GWAS trait on the molecular trait[20]) or horizontal pleiotropy (i.e., the identified shared genetic variant drives the molecular and trait perturbation independently). In comparison, MR studies, which are conceptually similar to TWAS, use multiple genetic variants as instrumental variables (IVs) and are less prone to reverse causality and artefacts arising from LD patterns[21] - although horizontal pleiotropy can never be ruled out entirely. In addition, MR analyses allow the quantification - direction and magnitude - of the causal effect of the omic on the outcome trait.

With the advent of QTL datasets with increased sample sizes[6,8], opportunities to integrate GWAS data with multiple molecular traits are no longer hampered by low statistical power. Previous efforts integrating multiple QTL omics data either adopted colocalization strategies[22,23] or combined pairwise MR associations (two-step MR)[24,25] testing only a single molecular mediator. Multivariable MR (MVMR) approaches have been proposed to identify multiple mediators of

[1]University Center for Primary Care and Public Health, Lausanne, Switzerland. [2]Swiss Institute of Bioinformatics, Lausanne, Switzerland. [3]Department of Computational Biology, University of Lausanne, Lausanne, Switzerland. [4]Center for Integrative Genomics, University of Lausanne, Lausanne, Switzerland. [5]These authors jointly supervised this work: Eleonora Porcu, Zoltán Kutalik. ✉e-mail: marie.sadler@unil.ch; zoltan.kutalik@unil.ch

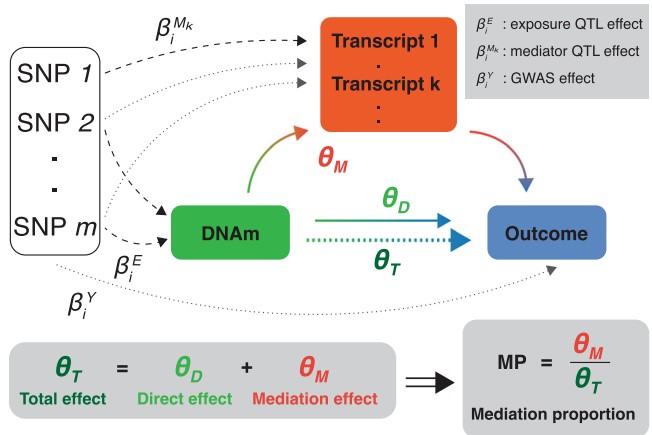

**Fig. 1 | Overview of the three-sample multivariable Mendelian randomization (3S-MVMR) design to quantify mediation of complex traits through DNA methylation (DNAm) and transcripts.** Genetic instruments (SNPs) are selected to be directly and significantly associated (dashed arrows) with either the exposure (DNAm, green) or any mediator $k$ (transcript in *cis*, red). The total effect $\theta_T$ (green-blue dotted arrow) of the exposure on the outcome (complex trait, blue) is estimated in a univariable MR analysis based on exposure-associated SNPs only. The direct effect $\theta_D$ (green-blue arrow) is estimated in an MVMR analysis on all valid instruments. The mediation effect $\theta_M$ (green-red and red-blue arrows) results from the difference between $\theta_T$ and $\theta_D$, and allows to calculate the mediation proportion (MP). The genetic effect sizes $\beta$ on the exposure, mediator and outcome come from m/eQTL and GWAS summary statistics, respectively. Transcripts were required to be causally associated to the DNAm-exposure to be included as mediators.

exposure-outcome relationships[26,27]. These approaches enable the dissection of the total causal effect of an exposure on an outcome into a direct and indirect effect measured via mediators. Similar to classical MR approaches, the use of genetic instruments allows for robust causal inference and MVMR has proven to be an unbiased approach for mediation analyses, even in the presence of confounders[26,27]. Hence, in addition to identifying causal effects through multiple layers, MVMR allows the quantification of mediation effects.

Here, we propose a three-sample MVMR (3S-MVMR) framework to quantify the role of *cis*-transcripts in mediating DNAm → complex trait causal relationships (Fig. 1). To do so we integrated methylation and transcript QTLs (mQTLs and eQTLs, respectively) with GWAS summary data of 50 clinically relevant traits to estimate global mediation proportions (MPs), i.e., the proportion of transcript-mediated causal effect relative to the total effect of DNAm on complex traits. In contrast with previous multi-omics integration methods, each 3S-MVMR regression analysis makes use of at least 5 near-independent instrumental variables (IVs) allowing for more robust causal inference and post-hoc sensitivity analyses. We performed simulation studies to assess biases of the 3S-MVMR estimates for MP under various parameter settings. In addition to quantifying the regulatory connectivity between DNAm and transcript levels, we investigated underlying factors driving high MPs, and hypothesized several mechanistic pathways between DNAm, gene expression and complex traits.

## Results

### Overview of the methods
We performed univariable and multivariable MR to estimate total ($\hat{\theta}_T$) and direct ($\hat{\theta}_D$) causal effects, respectively, with MP (mediation proportion) estimates being calculated as the ratio of the indirect effect (i.e., mediated through the molecular mediators) to the total effect of the exposure on the outcome trait[28] (Fig. 1; Eqs. (1) and (3)). If weak genetic instruments can introduce a bias towards the null in a univariable MR setting[29], this bias can be in any direction for MVMR studies[30]. Both sample size and choice of instruments and mediators

can introduce a bias in any direction[30], leading to under- or overestimations of the MP. To quantify these biases and assess the sensitivity of estimated $\widehat{MP}$s, we conducted simulation studies mimicking settings that emerge from real data applications (Methods; Supplementary Fig. 4).

We then applied our framework in a genome-wide screen to estimate $\hat{\theta}_T$ of DNAm sites on 50 outcomes and contrasted them to the effects not mediated by transcripts in *cis* ($\hat{\theta}_D$). Genetic effect sizes on the DNAm and transcript levels came from the largest publicly available mQTL and eQTL datasets, respectively, derived from whole blood[6,8]. MP estimates were then computed only for DNAm-trait pairs with significant Bonferroni-corrected $\hat{\theta}_T$ effects, grouped by trait, trait category and all pairs combined. We present MP results for DNAm-trait pairs with at least one mediator significantly associated to the exposure ("detectable mediation"), but also for pairs, including the ones without a significant causal effect on any potential transcript ("overall mediation"). The overall MP quantifies more accurately the role of *cis*-transcripts in mediating DNAm effects, as the restriction to only DNAm-trait pairs with a mediator could introduce a selection bias towards higher MPs. Additionally, we performed various sensitivity analyses on these MR results to assess the robustness of the MP estimates: assessing weak instruments (through conditional F-statistics), heterogeneity tests (through heterogeneity Q-statistics and leaving the strongest instrument out) and estimating bias due to by-chance signal overlap (through simulations).

### Simulation results
We performed simulation studies to assess the bias in estimated MPs ($\widehat{MP}$) by exploring a wide range of realistic parameter settings which cover at least the interquartile range as observed in real data (Supplementary Figs. 4-5; Supplementary Tables 1-2; Methods). Using default settings (i.e., median values for each parameter such as 2 true mediators $N_{med}$ and a true MP of 35%), the bias in $\widehat{MP}$ is minimal with the mean $\widehat{MP}$ equalling 33.5% (95% CI: [32.0%−35.0%]; Supplementary Fig. 6; Supplementary Table 2). A determining factor in accurately estimating MPs was the available sample size to derive the mediator QTL effects. Low sample sizes resulted in significant underestimations of the MP, with mediator sample size of 3000 compared to 30,000 resulting in a 17% relative decrease (6% in absolute values) of the estimated $\widehat{MP}$ (Fig. 2a). The reason for this significant underestimation was not only weak instrument bias, but also the omission of relevant mediators with on average only 1.17 ($N_{med,sig}$) out of the 2 ($N_{med}$) relevant mediators detected at a sample size of 3000 (Fig. 2a). We further tested the robustness of the $\widehat{MP}$ with respect to the number of included mediators by varying the mediator selection threshold $P_{EM}$. Among a set of 20 potential mediators, those not passing the $P_{EM}$ as determined by univariable MR effects of the exposure on each of these mediators were excluded from the MVMR model (Methods). Using a too lenient or too stringent $P_{EM}$ threshold resulted in downward biased $\widehat{MP}$s (Fig. 2b), as the former leads to the inclusion of too many non-mediators in the model (giving rise to weak instrument bias), while the latter case fails to include relevant mediators in the model. The used mQTL and eQTL datasets provide SNP effect sizes in *cis* of the assessed DNAm probe and transcript levels, respectively, and were primarily restricted to significant mQTLs for the former. Thus, in the MVMR analysis SNP-exposure effects for mediator instruments are often non-significant (hence unreported) and set to zero to reduce regression dilution bias (i.e., weak instrument bias). Our simulation studies, which mimicked this scenario by setting non-significant effects to zero (Methods), confirmed that this did not introduce any bias.

Furthermore, we investigated weak instrument bias of both exposure- and mediator-associated IVs. When mediator-associated IVs were weak (i.e., low direct mediator heritabilities ($h^2_{M,direct}$; Methods), a high variability and significant underestimation of the $\widehat{MP}$ was observed (Fig. 2c). In case of low mediator heritability, the conditional

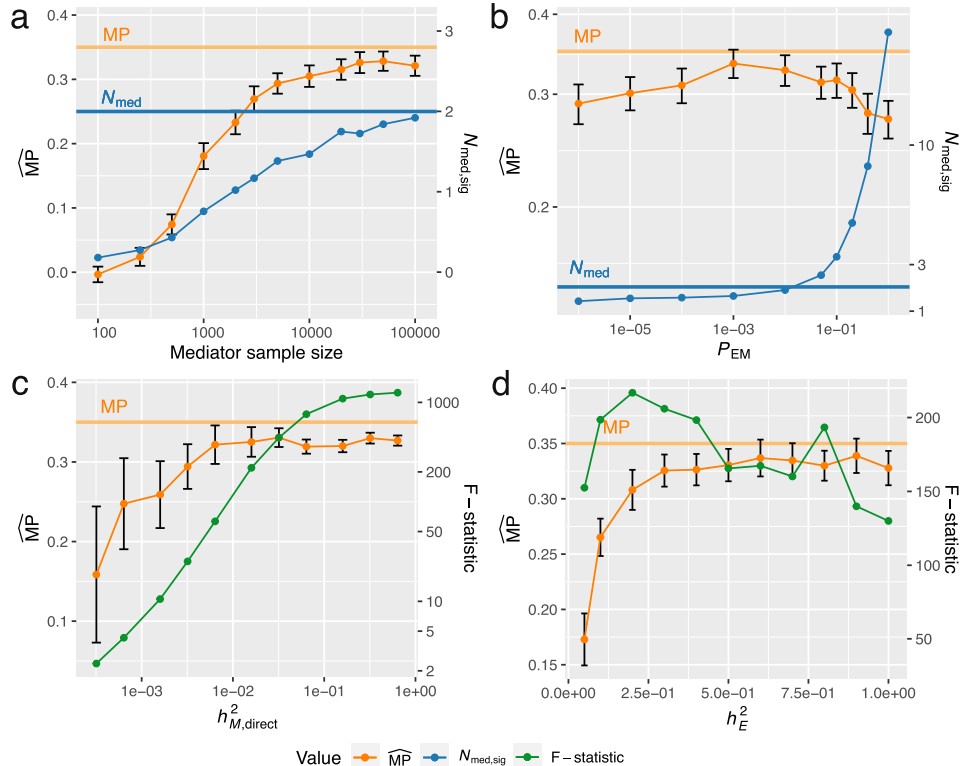

**Fig. 2 | Simulation results to assess the bias in estimated mediation proportions ($\widehat{\text{MP}}$s) in real data settings. a** Influence of the mediator sample size on the estimated $\widehat{\text{MP}}$ (orange) and number of selected mediators ($N_{\text{med,sig}}$, blue). **b** Influence of the mediator selection threshold $P_{\text{EM}}$. **c, d** Sensitivity of $\widehat{\text{MP}}$s in settings of weak instruments, simulated by low mediator ($h^2_{M,\text{direct}}$) and exposure ($h^2_E$) cis-heritabilities, respectively. Conditional F-statistics (green) of the exposure allow to test for weak instrument bias (critical values are defined at a threshold of F-statistic < 10). For a given parameter setting, 500 exposure-outcome pairs were simulated on which an $\widehat{\text{MP}}$ (orange points) and 95% CI (black error bars) were estimated. The true MP of the model was 0.35 (horizontal orange lines), and the true number of relevant mediators $N_{\text{med}}$ was 2 (horizontal blue lines) among a set of 12 potential mediators (20 in **b**).

F-statistics of the exposure was also below the critical threshold of < 10 (Methods) indicating weak instruments. Similarly, for low exposure heritability ($h^2_E$), underestimated $\widehat{\text{MP}}$s were obtained, even in case of high conditional F-statistics ( >120; Fig. 2d). Additional simulation studies with more polygenic exposures and increased number of relevant mediators $N_{\text{med}}$ for different exposure and mediator heritabilities corroborated the findings of underestimated $\widehat{\text{MP}}$s in case of weak instruments (Supplementary Fig. 7).

**Application to 50 complex traits**
We first estimated the causal effects of DNAm probes on 50 complex traits, ranging from biomarkers indicative for diseases, such as low-density lipoprotein (LDL) and glucose levels, to diseases such as asthma and schizophrenia (Supplementary Data 1). DNAm-trait pairs with a significant total causal MR effect ($P_T < 1e{-}6$) were then further assessed to examine what fraction of the DNAm → trait causal effect is mediated by transcripts in *cis* (Fig. 3a; Supplementary Fig. 1). Mediation analyses could be conducted for 2069 pairs, for which at least 1 transcript was causally associated to the DNAm exposure (detectable mediation). First, we regressed $\hat{\theta}_D$ against $\hat{\theta}_T$ within each trait influenced by at least 10 DNAm probes while accounting for regression dilution bias[31] (Eq. (6)). $\widehat{\text{MP}}$s estimated for each of these 41 traits ranged from 18.0 to 78.0% (mean: 36.9%, 95% CI: [13.5%–60.3%]) with the trait with the highest $\widehat{\text{MP}}$ being grip strength and the one with the lowest testosterone level (Fig. 3b). Regressing $\hat{\theta}_D$ against $\hat{\theta}_T$ for all pairs combined yielded an $\widehat{\text{MP}}$ of 37.8% (95% CI: [36.0%–39.5%]) (Fig. 3c). Grouping the traits into 10 physiological categories (Supplementary Data 1) showed that the $\widehat{\text{MP}}$ was highest for hepatic biomarkers (mean: 46.6%, 95%CI: [41.5%–51.7%]), followed by renal biomarkers (mean:

43.5%, 95%CI: [37.5%–49.5%]). In contrast, adiposity-related and hormonal traits exhibited the lowest $\widehat{\text{MP}}$ (Fig. 3b; Supplementary Fig. 8).

In addition to the 2069 DNAm-trait pairs with detectable mediation, there were 554 pairs testable for mediation, but with no detectable causally implicated transcript (Fig. 3a). Setting $\hat{\theta}_D$ to $\hat{\theta}_T$ for these pairs and regressing $\hat{\theta}_D$ against $\hat{\theta}_T$ for all 2623 DNAm-trait pairs combined reduced the $\widehat{\text{MP}}$ to 28.3% (95% CI: [26.9%–29.8%]) (Fig. 3d). We refer to this $\widehat{\text{MP}}$ as the overall $\widehat{\text{MP}}$, as it is a more objective measure of the importance of the transcriptome in mediating DNAm-to-phenotype effects. While more reflective of mediated DNAm effects, it may also be overly conservative since the set of testable transcript mediators ($N = 19,250^8$) is a magnitude lower than that of the whole transcriptome[32].

The average number of mediator transcripts, potentially correlated, was 3.3 per methylation-trait pair with detectable mediation, indicating that the impact of methylation is not mediated by a single transcript. To further explore this observation, we assessed the extent to which DNAm → trait effects were mediated by the single most significantly DNAm-associated transcript ("top" transcript; Methods), as opposed to all transcripts in *cis*. This resulted in an $\widehat{\text{MP}}_{\text{top}}$ of 26.0% (range: [13.0%–46.8%]) averaged across the 41 traits, and an $\widehat{\text{MP}}_{\text{top}}$ of 26.6% (95% CI: [25.1%–28.1%]) when aggregating the 2069 DNAm-trait pairs (Supplementary Fig. 9). This significant drop in the $\widehat{\text{MP}}$ ($P_{\text{diff}} < 5e{-}21$) corroborates our initial hypothesis that DNAm sites regulate the expression of multiple transcripts in the *cis* region.

**MVMR sensitivity analyses**
We conducted MVMR sensitivity analyses to assess potential sources of bias of the MP estimates such as weak instruments and pleiotropy.

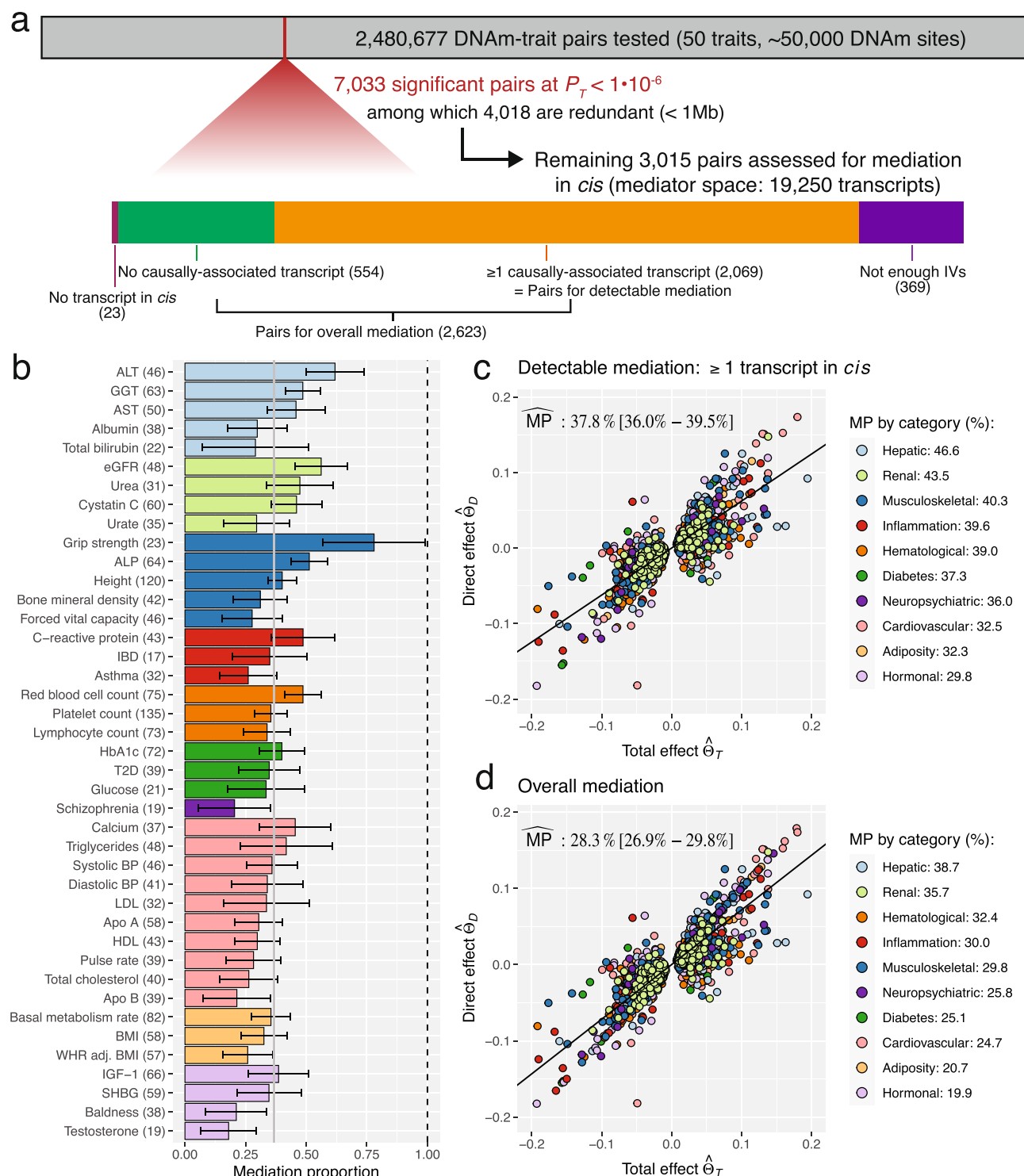

**Fig. 3 | Mediation proportions ($\widehat{MP}$s) for transcripts in *cis* mediating DNAm-to-trait effects. a** Flowchart describing the selection of DNAm-trait pairs retained for mediation analyses. Among the total of 2,480,677 pairs tested, 2069 pairs (orange) with a significant total causal effect ($P_T$) and at least 1 causally-associated transcript were assessed for mediation (**b**, **c**). 554 pairs without any transcript causally linked to the DNAm site (green) were included in the calculation of the overall mediation. Pairs without any transcript in the *cis* region (pink) were omitted in mediation analyses as were pairs without sufficient instrumental variables (IVs, purple). **b** $\widehat{MP}$s by trait where error bars denote the 95% CI, and the grey vertical bar shows the mean $\widehat{MP}$ across the traits ($\widehat{MP}$s per trait were derived by regressing $\hat{\theta}_D$ against $\hat{\theta}_T$).

Only traits with ≥10 DNAm-trait pairs with detectable mediation are displayed (41 traits with number of evaluated pairs indicated in parentheses), colour-coded by their physiological category as defined in the legends of **c** and **d**. **c** Detectable mediation: All DNAm-trait pairs assessed in the mediation analyses (2069) with traits being grouped into 10 physiological categories. The global $\widehat{MP}$ in percentage with 95% CI is shown in the plotting area and individual category $\widehat{MP}$s in the legend. **d** Overall mediation: Same analysis as in **c**, but including all 2623 DNAm-trait pairs with at least 1 transcript present in the *cis* region. For these additional pairs, the direct effect was set to the total effect.

To test whether the MVMR estimates suffer from weak instrument bias, we calculated conditional F-statistics[33]. These statistics reflect whether genetic variants sufficiently explain the variance in the exposure given the presence of mediators. As demonstrated by Sanderson et al., direct effect estimates ($\hat{\theta}_D$) of exposure-trait pairs for which the F-statistic is ≤10 might be biased[33]. Among the 2069 DNAm-trait pairs, 1061 had an F-statistic > 10 with an $\widehat{MP}$ of 35.5% (95% CI: [33.6%–37.5%]) which was not significantly lower than the one for all pairs combined ($P_{diff}$=0.09). Pairs with F-statistics ≤10 (N=1008) had significantly more mediators (4.32 vs 2.35, two-sided $t$-test: $P$=2.13e-64), but not a significantly higher $\widehat{MP}$ (mean: 40.9%, 95% CI: [37.8%–44.0%]; $P_{diff}$= 0.08; Supplementary Figs. 10-11).

Pleiotropic IVs violate MR assumptions and heterogeneity tests, such as the Cochran's Q-statistic, can be used to detect them, assuming that most IVs are valid[34]. We calculated Q-statistics for the IV sets in both the univariable and multivariable MR analyses. Out of the 2069 DNAm-trait pairs, 1757 showed no signs of heterogeneity in the univariable MR analyses ($P_{HET}$ > 0.01) and 1405 in neither the univariable nor multivariable analyses. The $\widehat{MP}$ of these 1405 pairs was not significantly different from the overall one (mean: 38.3%, 95% CI: [36.1%–40.6%]; $P_{diff}$=0.7; Supplementary Fig. 12).

Next, we assessed the influence of the $p$-value threshold $P_{EM}$ to select mediators based on the exposure-to-mediator causal effect (default $P_{EM}$=0.01 for which N=2069 DNAm-trait pairs with at least 1 mediator were found). With a more lenient threshold ($P_{EM}$=0.05), more DNAm-trait pairs with mediators emerged (N=2189). Conversely, with a more stringent threshold ($P_{EM}$=0.001), less pairs were detected (N=1881). No differences in MPs between the three settings were found in these detectable mediation analyses ($P_{diff}$ > 0.05; Supplementary Fig. 13), but when calculating the overall MP (i.e., inclusion of all DNAm-trait pairs with potential transcript mediators in the $cis$-region) on a common set of DNAm-trait pairs (N=2543, $\widehat{MP}_{overall,P01}$=27.6% (95% CI: [26.1%–29.2%])), a significantly higher MP for the more lenient threshold ($\widehat{MP}_{overall,P05}$=32.0% (95% CI: [30.4%–33.6%]); $P_{diff}$=1.1e-4), and significantly lower MP for the more stringent threshold were observed ($\widehat{MP}_{overall,P001}$=24.6% (95% CI: [23.2%–26.1%]); $P_{diff}$=4.8e-3; Supplementary Fig. 14).

Finally, we conducted sensitivity analyses to determine whether significant MR associations were due to horizontal pleiotropy. Regulatory pathways between DNAm exposure probes and transcript mediators were assessed in $cis$. As such, SNPs in LD with significant QTLs for both quantities could give rise to an association merely because of horizontal pleiotropy (i.e., due to random overlap between $cis$-QTLs in close vicinity), an issue further exacerbated by the fact that molecular omics entities generally have fewer associated IVs than complex traits. To assess whether mediation results are only based on a single strong genetic instrument, we repeated the mediation analysis excluding the top IV (i.e., exposure-associated IV with the lowest $p$-value) from both the total effect $\theta_T$ and direct effect $\theta_D$ calculations (Methods). The results show that while MR effect estimates remain concordant in magnitude and effect direction, the estimates are noisier due to the much weaker instruments (significantly lower F-statistics; two-sided $t$-test: $P$=5.37e-11; Supplementary Fig. 15). MP estimates were also higher when the top IV was excluded ($\widehat{MP}$ =47.3% (95% CI: [38.4%–56.2%]); $P_{diff}$=0.023; Supplementary Fig. 15), however, this was no longer the case when controlling for conditional F-statistics > 10 ($\widehat{MP}$ =40.9% (95% CI: [29.3%–52.4%]); $P_{diff}$=0.48). Additionally, we performed simulation analyses to assess the possibility of significant DNAm-transcript associations caused by $cis$-mQTL and -eQTL signals being in LD (Methods). The analysis shows that randomly picked eQTL-SNPs in the region result in slightly inflated, but much weaker MR associations than using the original eQTL data (Supplementary Figs. 16-17). The results indicate that by-chance LD between $cis$-mQTLs and -eQTLs can yield false positive findings, but those signals are

substantially weaker than the ones observed in real data. In other words, mQTL and eQTL IVs are in much higher LD than expected by chance.

Overall, these sensitivity analyses showed that the estimated MPs remain robust when removing DNAm-trait pairs that potentially violate MVMR assumptions, while also suggesting that the set $P_{EM}$ threshold of 0.01 may lead to underestimated MP estimates. Finally, we found strong evidence that molecular associations mediating DNAm-trait effects are predominantly due to vertical pleiotropy, even when only a limited number of IVs were available.

### Determining factors of mediation proportions

We explored underlying factors driving high MPs through transcript levels (Fig. 4a). $\widehat{MP}_{top}$ decreased with increased distances between the DNAm site and the gene transcription start site (TSS) of the top transcript ($\rho = -0.076$, $P = 5.2$e-4; Fig. 4b). This distance was also negatively correlated to the DNAm-to-transcript MR squared effect size, $\alpha_{EM}^2$, ($\rho = -0.13$, $P = 3.1$e-19; Fig. 4c), which in turn was a good predictor for high MPs ($\rho = 0.39$, $P = 2.5$e-75; Fig. 4d). The mediation proportion was the highest for DNAm sites residing in the first exon, followed by those in the 5′UTR, within 200 bp of the TSS and lowest for those within 1500 bp of the TSS and in the gene body (Supplementary Fig. 18).

DNAm inhibiting the binding of transcription factors (TFs) thereby repressing gene expression is often alluded to as the classical mechanism of action for DNAm[35]. From the 1,066,307 unique DNAm-to-transcript causal effects assessed, 47,445 were significant at $P < 4.7$e-8. Although negative effects had a larger magnitude than positive ones (two-sided $t$-test: $P = 0.0082$) only 53.4% of DNAm → transcript causal effects were negative. Stratifying DNAm sites with respect to their location on the assessed transcript, we found that DNAm sites situated in the first exon and nearby the TSS were enriched for negative effects ($P = 2.7$e-3, 1.2e-5 and 3.8e-4 for 1st exon, TSS ± 1500 bp and TSS ± 200 bp, respectively), whereas those in the gene body were enriched for positive ones ($P = 2.2$e-10; Supplementary Table 3). These observations are in line with previous studies that only showed a slight trend for negative methylation-gene expression correlations[36–39]. We further tested whether the MR DNAm-to-transcript causal effects correlated with reported methylation-transcript correlations[37] and found a strong agreement ($\rho = 0.39$, $P = 2.6$e-18, 471 DNAm-transcript pairs).

Consistent with higher MPs when mediating through multiple transcripts, we found a strong correlation between the number of mediators and the MP ($\rho = 0.39$, $P = 4.4$e-75; Fig. 4e). Many of these mediators were correlated amongst each other, which in theory should be accounted for by the MVMR model. To ensure that this was the case, we repeated the mediation analysis with uncorrelated mediators ($R_{med} < 0.3$; Methods). The mean number of selected mediators dropped by more than half, from 3.3 to 1.2 (Supplementary Fig. 19), and the $\widehat{MP}$ across all DNAm-trait pairs decreased ($\widehat{MP}_{uncorrelated}$ = 30.5% (95% CI: [28.8%–32.1%])), while remaining significantly higher than $\widehat{MP}_{top}$ ($P_{diff}$ = 6.6e-4). Decreasing the $R_{med}$ threshold to 0.2 and 0.1 did not significantly decrease $\widehat{MP}_{uncorrelated}$ ($P_{diff}$ > 0.05), which stabilized at 29.2% (95% CI: [27.5%–30.8%]) for $R_{med} < 0.1$ (Supplementary Fig. 19).

Furthermore, we investigated whether $\widehat{MP}$s are dependent on the DNAm → transcript causal effect directions following the logic of a recent DNAm-transcript correlation study[39]. To this end, we stratified DNAm-trait pairs by the $\alpha_{EM}$ sign and number of mediators (Table 1). If there was only a single mediator, $\widehat{MP}$s were significantly higher if the DNAm was decreasing expression ($P_{diff}$ = 3.49e-8). This is consistent with the observation that negative effects $\alpha_{EM}$ were larger than positive ones and the positive correlation between $\alpha_{EM}$ magnitudes and high $\widehat{MP}$s (Fig. 4d). When there were multiple mediators, most DNAm sites had negative effects on some transcripts and positive effects on others. These bivalent DNAm probes exhibited the highest $\widehat{MP}$s

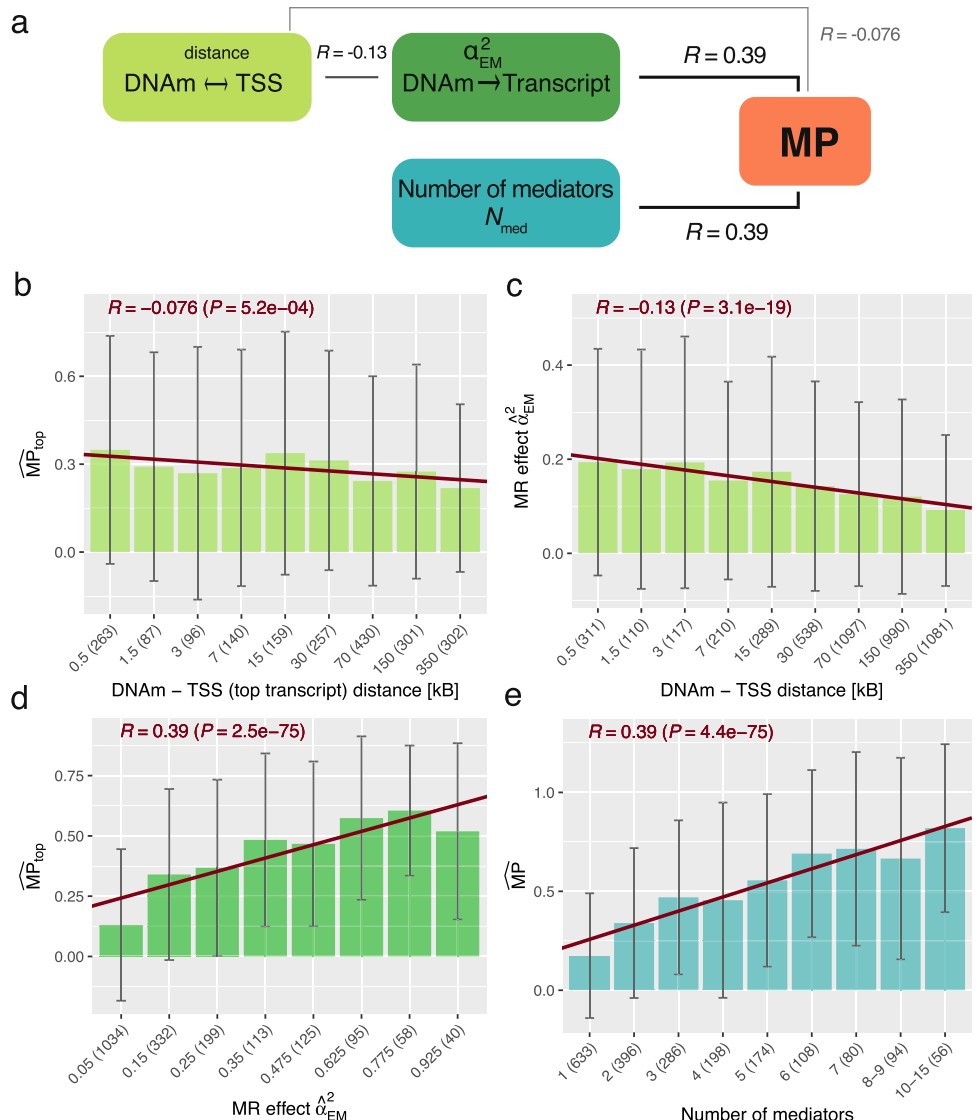

**Fig. 4 | Exposure-to-mediator regulatory strength and number of mediators explaining mediation proportions (MPs). a** Summary of the correlations ($R$) between MP (red) and DNA methylation (DNAm)-to-transcript causal MR effects ($\hat{\alpha}_{EM}^2$, dark green), distance between the DNAm site and transcription start site (TSS, light green) and number of mediators ($N_{med}$, blue). **b** Average MP through top transcript ($\widehat{MP}_{top}$) of DNAm-trait pairs stratified according to the distance between the DNAm site and the TSS of the top transcript. All DNAm-trait pairs with at least one mediator were included (2069 pairs). **c** Average MR causal effects ($\hat{\alpha}_{EM}^2$) of DNAm-transcript pairs stratified according to the distance between the DNAm site and the TSS. Unique DNAm-transcript mediator pairs across all DNAm-trait pairs were included (4743 pairs). **d** Average $\widehat{MP}_{top}$ of DNAm-trait pairs stratified according to DNAm-to-top transcript MR causal effect size $\hat{\alpha}_{EM}^2$. All DNAm-trait pairs with at least one mediator were included (2069 pairs). **e** Average $\widehat{MP}$ of DNAm-trait pairs stratified according to the number of mediators. All DNAm-trait pairs with at least one mediator were included (2069 pairs). The reported $p$-values ($P$) for the corresponding Pearson correlations ($R$) arise from a two-sided t-test and were calculated between the two respective quantities on DNAm-trait/DNAm-transcript pairs prior stratification. Bin height represents mean within each bin and the error bars the corresponding standard deviations (number of evaluated pairs within each bin is indicated in parentheses). The red slope represents the regression fit between the bin's positions and heights, and serves merely for visualization purposes.

($\widehat{MP} = 53.9\%$ (95% CI: [51.2%−56.5%])) - a consequence of being causally associated to more mediators than average (5.01 vs 3.31), with $N_{med}$ being a strong predictor for high $\widehat{MP}$s (Fig. 4e). Combining DNAm-trait pairs with single and multiple mediators, but with consistent negative or positive $\alpha_{EM}$ values, the observation of higher $\widehat{MP}$s when DNAm was decreasing transcript levels persisted ($P_{diff} = 0.020$).

### Putative regulatory mechanisms of action

In addition to providing insights into global patterns governing the mediation between different intermediate phenotypic layers and functional traits, our analyses generated plausible hypotheses regarding specific biological pathways. We chose to follow-up putative regulatory mechanisms of DNAm-to-complex traits through transcript

levels which showed both strong total effects ($|\hat{\theta}_T|>0.02$) and substantial mediation proportion ($\widehat{MP}>0.2$; complete list in Supplementary Data 2).

Involvement of the anti-oxidant and anti-inflammatory protein PARK7 in inflammatory bowel disease (IBD) has recently been brought to light[40–43]. While the exact role of the protein in the disease remains debated, reduced intestinal expression of *PARK7* was observed in patients and mouse models for IBD[43]. Moreover, *Park7* knockout mice were shown to have increased levels of pro-colitis bacterial species in their microbiome[42,44] and experience aggravated symptoms of experimentally-induced colitis[43]. In line with these observations, DNAm of the *PARK7* promoter probe cg10385390 (chr1:8'022'505) decreased *PARK7* transcript expression ($\hat{\alpha}_{EM} = -0.675$, $P = 2.7e-4$;

**Table 1 | Exposure-to-mediator effect direction and number of mediators explaining MPs**

| | Negative | Positive | Bivalent | Total |
|---|---|---|---|---|
| **Mono** | $N$ = 370 (17.9%) | $N$ = 276 (13.3%) | 0, by definition | $N$ = 646 (31.2%) |
| | $\widehat{MP}$ =20.8% | $\widehat{MP}$ =7.97% | | $\widehat{MP}$ =14.7% |
| | (95% CI: [17.3%–24.2%]) | (95% CI: [5.04%–10.9%]) | | (95% CI: [12.3%–17.1%]) |
| **Multi** | $N$ = 239 (11.6%) | $N$ = 207 (10.0%) | $N$ = 977 (47.2%) | $N$ = 1423 (68.8%) |
| | $\widehat{MP}$ =42.8% | $\widehat{MP}$ =42.8% | $\widehat{MP}$ =53.9% | $\widehat{MP}$ =50.3% |
| | (95% CI: [38.3%–47.2%]), | (95% CI: [38.3%–47.3%]), | (95% CI: [51.2%–56.5%]), | (95% CI: [48.2%–52.4%]), |
| | mean ($N_{med}$)=3.01) | mean ($N_{med}$)=2.84) | mean ($N_{med}$)=5.01) | mean ($N_{med}$)=4.35) |
| **Total** | $N$=609 (29.4%) | $N$ = 483 (23.3%) | $N$ = 977 (47.2%) | $N$=2069 (100%) |
| | $\widehat{MP}$ =27.7% | $\widehat{MP}$ =22.8% | $\widehat{MP}$ =53.9% | $\widehat{MP}$ =37.8% |
| | (95% CI: [24.8%–30.5%]), | (95% CI: [19.8%–25.8%]), | (95% CI: [51.2%–56.5%]), | (95% CI: [36.0%–39.5%]), |
| | mean ($N_{med}$)=1.79) | mean ($N_{med}$)=1.79) | mean ($N_{med}$)=5.01) | mean ($N_{med}$)=3.31) |

DNAm-trait pairs were stratified by the number of mediators ($N_{med}$; "mono" if $N_{med}$ =1 and "multi" if $N_{med}$ > 1) and by the exposure-to-mediator $\alpha_{EM}$ causal effect sign ("Negative" and "Positive" for DNAm decreasing and increasing transcript levels, respectively, and "Bivalent" if a given DNAm site is affecting transcript levels in both directions. For each stratum, the number of DNAm-trait pairs ($N$), the estimated mediation proportion ($\widehat{MP}$) and mean $N_{med}$ is shown.

Fig. 5a). High transcript levels decrease IBD risk ($\hat{\alpha}_{MY} = -0.131$, $P = 1.7e$-7) resulting in an overall increased IBD risk upon DNAm ($\hat{\theta}_T = 0.114$, $P = 8.2e$-9).

Despite often being associated with decreased expression[35], our data provides examples of methylation boosting expression. For instance, DNAm of cg13428477 (chr3:122'748'086) increased *PDIA5* expression ($\hat{\alpha}_{EM} = 0.333$, $P = 7.3e$-11), whose levels subsequently increased platelet count ($\hat{\alpha}_{MY}$ =0.062, $P = 0.018$), so that DNAm resulted in significantly increased platelet count ($\hat{\theta}_T = 0.056$, $P = 1.3e$-43) (Fig. 5b). Association between the *PDIA5* locus and platelet count was reported through GWAS[45]. Platelets are small cell fragments produced by megakaryocytes, which themselves are derived from hematopoietic stem cells. Accordingly, *PDIA5* has a binding site for the hematopoietic stem and progenitor cell TF MEIS1[46] and is overexpressed in megakaryocytes as compared to other blood cell types[47]. Further studies showed that *pdia5* protein knockdown in zebrafish resulted in strongly decreased platelet count[48], matching our findings and confirming the role of *PDIA5* in thrombopoiesis.

In another example, we observed that DNAm of cg09070378 (chr1:161'183'762) decreased asthma risk ($\hat{\theta}_T = -0.031$, $P = 8.1e$-11) by reducing *FCER1G* expression ($\hat{\alpha}_{EM} = -1.0$, $P = 3.5e$-18), a gene listed in the KEGG pathway for asthma (hsa05310) and whose expression associated with an increased risk for asthma ($\hat{\alpha}_{MY} = 0.019$, $P = 3e$-12) (Supplementary Fig. 21). The *FCER1G* promoter was found to be hypomethylated in patients with atopic dermatitis, with DNAm levels correlating negatively with the gene's expression[49], suggesting a broad role of *FCER1G* in allergic disorders. Our data also supports and provides a mechanistic explanation for the recent finding that reduced *IFNAR2* expression causally decreases the odds of severe coronavirus disease 2019 (COVID-19)[50,51], which was later supported by the increased susceptibility for severe COVID-19 in individuals with rare loss-of-function mutations in *IFNAR2*[52]. Indeed, we found that DNAm of the *IFNAR2* promoter probe cg13208562 (chr21:34'603'264) decreased the gene's expression ($\hat{\alpha}_{EM} = -0.446$, $P = 2.4e$-19) (Supplementary Fig. 22). As *IFNAR2* expression protects against hospitalization following COVID-19 infection ($\hat{\alpha}_{MY} = -0.090$, $P = 4.2e$-6), DNAm of the locus increased the risk of severe infection ($\hat{\theta}_T = 0.064$, $P = 8.5e$-13).

## Discussion

We presented a framework to quantify mediation of complex trait-impacting effects through an omics layer and demonstrated its application to assayed blood-derived DNAm (exposure) and transcript levels (as mediator). Evidence for mediation of DNAm-to-trait effects through transcripts in *cis* was found to be at least 28.3% for the 2623 DNAm-trait pairs with significant total causal effects that could be assessed. While many robust methods are available for univariable MR, it is not the case for MVMR[26,27]. Still, we could confirm the robustness of our MVMR estimates through various sensitivity analyses (conditional F-statistic, heterogeneity Q-statistic, excluding the strongest IV) that could not pinpoint any factor drastically biasing our MP estimates. Importantly, simulation studies indicated that MP estimates were likely to be lower bounds. Low sample size was shown to lead to MP underestimations, as do weak instruments, both for exposure- and mediator-associated IVs.

Additionally, we quantified the causal connectivity and directionality between DNAm and transcript levels and its impact on MPs. We found that 46.6% of significant DNAm-to-transcript effects were of positive sign (i.e., DNAm increasing transcription), particularly so when the DNAm site was situated in the gene body ($P_{Enrichment}$=2.15e-10). Interestingly, MPs were higher when DNAm was downregulating rather than upregulating transcripts. Previous genome-wide methylation and gene expression association studies reported high fractions of positive correlations (30–41%)[36,37,39] and further investigations indicated that our estimated methylation-to-transcript causal effects agree strongly with the respective correlations reported by Grundberg et al. ($P$=2.6e-18). While poorly understood[38], several mechanisms have been proposed to explain the phenomenon of DNAm induced transcription: preferential binding of some transcription factors to methylated DNA[53,54], prevention of repressor binding indirectly leading to increased expression through looping DNA[24,55], or DNAm in the gene body promoting elongation efficiency and preventing spurious initiation of transcription[56]. Furthermore, MP estimates indicated that DNAm sites typically regulate multiple transcripts in *cis* and that mediation through transcripts decreased the further away the TSS of the mediator transcript was from the DNAm site. Collectively, these results describe a more diverse picture of the transcription machinery, going beyond the classical views that DNAm solely reduces gene expression in the TSS region.

Statistical methods to integrate GWAS with omics data have seen a surge in recent years. Namely, colocalization methods based on a single genetic signal or corroborated by a secondary one, as well as methods supported by the SMR HEIDI statistic have been previously used in the study of DNAm-to-complex trait effects[6,14,24]. In the most recent publication of the GoDMC consortium, the former strategy was applied to systematically evaluate DNAm and GWAS co-localizing signals and compare them to MR[6]. This revealed a relatively poor overlap between colocalization and MR results, as both approaches have their weaknesses in detecting causal relationships. The major weakness of colocalization analysis is that it cannot detect directionality and does not estimate causal effect size. Colocalization of local association signals of two traits may be due to causal effects in either direction, common local confounder effect (e.g., shared regulatory mechanism) or causal markers in very high LD. Lack of colocalization can happen even if there is a true causal relationship, but there are additional associations impacting only the outcome trait. On the other hand, the major weakness of MR is that it may falsely detect a causal relationship when the causal variants for each trait are in reasonably high LD. The comparison of these two approaches is out of the scope of this work, but to explore the above-mentioned weakness in our study, we performed simulations tailored to detect by-chance overlaps in the association signals for methylation and gene expression (see pleiotropy sensitivity analyses for details). These analyses indicated that indeed elevated false positive rates are expected for MR, but the resulting MR

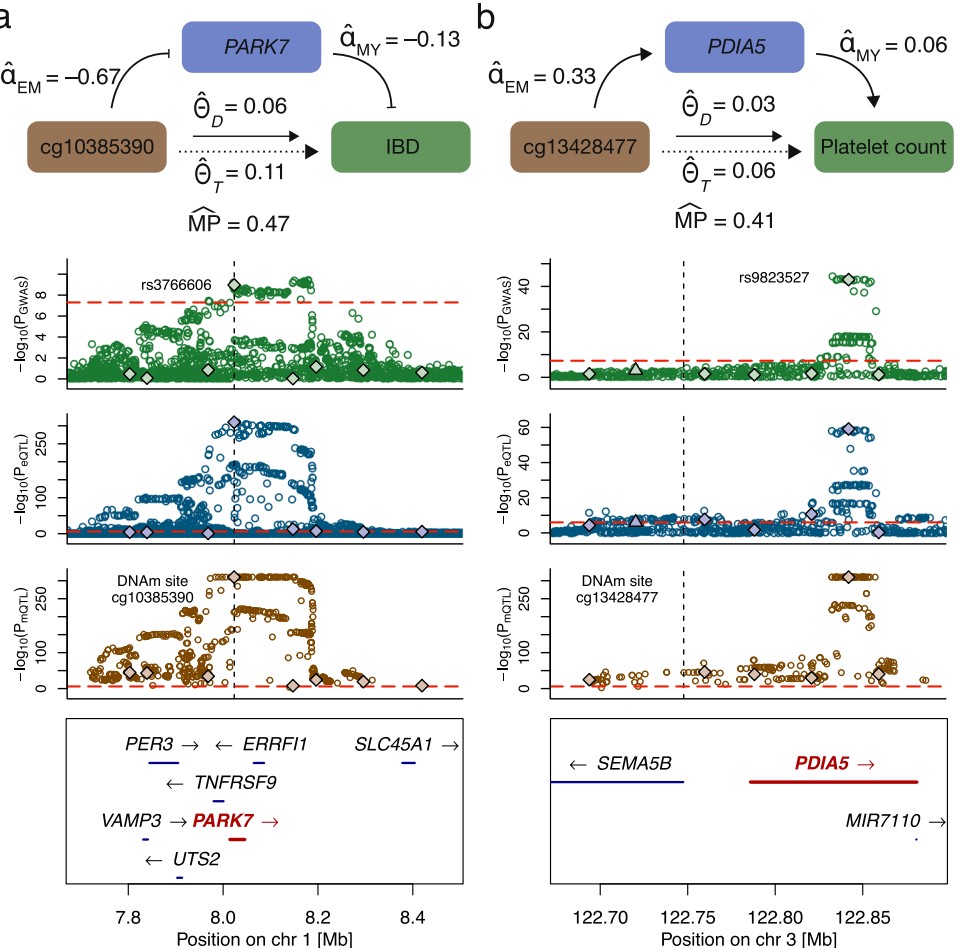

**Fig. 5 | Plausible DNAm-transcript-trait regulatory mechanisms. a** Mechanism involving DNAm probe cg10385390, *PARK7* and inflammatory bowel disease (IBD). **b** Mechanism involving DNAm probe cg13428477, *PDIA5* and platelet count. The top row displays a schematic of the mechanism with estimated univariable (total effect $\hat{\theta}_T$, DNAm-to-transcript effect $\hat{\alpha}_{EM}$ and transcript-to-outcome effect $\hat{\alpha}_{MY}$) and multivariable (direct effect $\hat{\theta}_D$) MR effects (displayed mediation proportions $\widehat{MP}$s) are derived from $\hat{\theta}_D$ and $\hat{\theta}_T$ estimates). The three following rows show the regional SNP associations ($-\log_{10}$(p-values)) with the trait (GWAS, green), transcript (eQTL,

blue) and DNAm (mQTL, brown) probe, respectively. Solid diamonds represent DNAm-associated instruments used in the univariable (for $\hat{\theta}_T$ calculation) and multivariable (for $\hat{\theta}_D$ calculation) MR analyses. Upwards pointing triangles are transcript-associated SNPs that were additionally included in the MVMR instrument set. Red dashed lines indicate the significance thresholds of the respective SNP associations. The vertical black dashed line represents the DNAm probe position. Bottom row illustrates the positions and strand direction of the genes in the locus.

p-values under the null are much less significant than the ones observed for real methylation-transcript data.

Mapping genetic variants identified by GWASs to biological processes is notoriously difficult[2]. In particular, a challenge in identifying causal chains through omics layers is the attenuation in the genetic association strengths when moving up along layers. In a linear model, the genetic effect on the phenotype is assumed to be the product of causal effects between the preceding layers and it was previously shown that the variance explained by the top associated QTL of the first layer weakens with each successive omics layer[24]. In line with this observation, the examples depicted in Fig. 5 visualize the decrease in the genetic associations from the DNAm to the complex trait level. While in the future our 3S-MVMR framework could be applied to further mediating layers (e.g. proteins or metabolites), current QTL datasets for these omics layers lack the dimensionality - both in terms of sample size and number of assessed entities. Once larger datasets become available, these could be used to support mechanistic findings resulting from transcript data.

While our method highlights candidate pathways and provides MP estimates, several limitations are to be considered. First, our MP estimates are based on a selection of 2623 DNAm-trait pairs with significant total effects ($P_T < 1e\text{-}6$), which inherently focuses on DNAm-trait pairs with larger (and hence detectable) effects. In theory, MPs could depend on the magnitude of the total causal effect, thus the reported MP may differ for weaker total effects. A special case of these weaker total effects is when direct and indirect effects differ in sign, leading to a weak total effect with an MP potentially outside the [0,1] range. Furthermore, selected DNAm sites were those with the strongest DNAm-trait signal in their region (up to 1Mb). Thus, we omit secondary methylation signals, which may be mediated by transcripts to a different degree. Second, as for all MR-IVW approaches, included IVs might be pleiotropic, i.e., violating MR assumptions and potentially biasing effect estimates. Although, filtering out DNAm-trait pairs with signs of heterogeneous IV sets did not change MP estimates, the presence of invalid IVs cannot be entirely excluded and could therefore compromise causal effect estimates[57,58]. In particular, since selected IVs are in *cis* of the investigated molecular trait, they might be based on a single (pleiotropic) haplotype signal. Third, we select mediators based on their association to the exposure without taking into account their mediator potential, i.e., whether or not the mediator is additionally causally linked to the trait. Phrased differently, selected mediators are simply candidates and such selection serves as a first filter to remove non-mediators. In line with our simulations, it has been shown that an extremely large number of such "false" mediators (88 out of 92) can

cause MVMR regression models to fail[30], indicating that our framework is less suitable for large numbers of molecular mediators unless the selection threshold $P_{EM}$ is made more stringent. Finally, while molecular mechanisms ought to be tissue- or even cell type-specific, QTL data used in this study were derived from whole blood. However, not correcting for blood cell types when analyzing gene expression data can introduce important artefacts[59]. It is also known that different tissues express different isoforms[60], with many splicing and expression QTLs shown to differ across tissues[61]. Accordingly, MPs for blood biomarkers were generally higher than those for diseases, for which blood might not be the most relevant tissue. Differences between biomarker and disease MPs might also be due to the fact that indirect pathways, through unmeasured mediators, play a greater role for the latter trait category. Once tissue-stratified multi-omics datasets of larger sample size become available, more accurate, and potentially higher MPs will be obtained in trait-relevant tissues.

To conclude, by adapting existing MVMR mediation techniques to molecular exposures and mediators, we quantified the causal connectivity between DNAm and transcript levels, and their importance in shaping complex traits. Overall, we found solid evidence that almost a third of DNAm-to-complex trait effects are mediated by transcripts in *cis*. Our integrative omics framework can be extended to other omics-GWAS combinations and provide a powerful tool for mapping GWAS signals to biological pathways and prioritizing functional follow-up experiments.

## Methods
### Univariable and multivariable Mendelian randomization
Univariable Mendelian randomization (MR) was applied to estimate the total causal effect ($\theta_T$) and multivariable MR (MVMR) to estimate the direct causal effect ($\theta_D$) of an exposure $E$ on an outcome $Y$. The mediation proportion (MP) was defined as $1 - \theta_D/\theta_T$. Under the MR assumptions, genetic variants $G$ used as IVs must be i) associated with $E$, ii) independent of any confounder of the $E - Y$ relationship, iii) conditionally independent of $Y$ given $E$. We analysed exposures with at least five LD-pruned ($r^2 < 0.05$) IVs associated ($P < 1e-6$) with the molecular exposure and located in *cis* ($<1$ Mb). To estimate $\theta_T$ we used the inverse-variance weighted (IVW) MR method, while accounting for (mildly) correlated instruments[19,62] as follows:

$$\hat{\theta}_T = \left(\boldsymbol{\beta}_E' \mathbf{C}^{-1} \boldsymbol{\beta}_E\right)^{-1} \boldsymbol{\beta}_E' \mathbf{C}^{-1} \boldsymbol{\beta}_Y \tag{1}$$

where $\boldsymbol{\beta}_E$ and $\boldsymbol{\beta}_Y$ are vectors of genetic effect sizes obtained from summary statistics for $E$ and $Y$, respectively. $\mathbf{C}$ is the linkage disequilibrium (LD) matrix with pairwise correlations between IVs estimated from the UK10K reference panel[63]. Sensitivity analyses confirmed that accounting for the LD-matrix safeguards against MR estimates being influenced by the pruning threshold $r^2$ (Supplementary Figs. 2-3). Since in the following MVMR model more IVs than mediators are required, we chose a more lenient pruning threshold ($r^2 < 0.05$), including IVs in mild LD (Supplementary Fig. 2). Prior to the causal effect calculations, IVs were Steiger-filtered to avoid that the IV's effect on $Y$ is significantly larger than it is on $E$[54] and were thus required to pass a threshold $t_{rev} < \frac{|\beta_{E_i}| - |\beta_{Y_i}|}{\sqrt{\text{var}(\beta_{E_i}) + \text{var}(\beta_{Y_i})}}$ with $t_{rev}$ set at $-2$, equivalent to a one sided test *p*-value threshold of 0.023[34]. IVs not passing this threshold are prone to violating the third MR assumption of horizontal pleiotropy since they are more directly linked to the outcome. As a result, MR estimates including such IVs would potentially mix up forward and reverse causal effects. The standard error (SE) of $\theta_T$ can be approximated by the Delta method[65]:

$$\text{SE}(\hat{\theta}_T) = \sqrt{\left(\boldsymbol{\beta}_E' \mathbf{C}^{-1} \boldsymbol{\beta}_E\right)^{-1} \boldsymbol{\beta}_E' \mathbf{C}^{-1/2} \boldsymbol{\Sigma} \mathbf{C}^{-1/2} \boldsymbol{\beta}_E \left(\boldsymbol{\beta}_E' \mathbf{C}^{-1} \boldsymbol{\beta}_E\right)^{-1}} \tag{2}$$

where $\boldsymbol{\Sigma}$ is a diagonal matrix with each diagonal element $i$ equalling the maximum of the regression variance $s^2$ and $\text{var}(\beta_{Y_i})$[34].

Through the inclusion of mediators $M_k$ and their associated *cis* genetic variants ($r^2 < 0.05$, $P < 1e-6$), $\theta_D$ can be estimated analogously to $\theta_T$ using a multivariable regression model[28] as the first element of $\boldsymbol{\theta}_D$:

$$\hat{\boldsymbol{\theta}}_D = \left(\boldsymbol{B}' \mathbf{C}^{-1} \boldsymbol{B}\right)^{-1} \boldsymbol{B}' \mathbf{C}^{-1} \boldsymbol{\beta}_Y \tag{3}$$

where $\boldsymbol{B}$ is a matrix with $k + 1$ columns containing the effect sizes of the IVs on the exposure in the first column and on each mediator in the subsequent columns. The remaining elements of $\boldsymbol{\theta}_D$ represent the direct effects of the mediators on the outcome and were referred to as $\alpha_{MY,k}$. In the estimation of MPs, we were not interested in $\alpha_{MY,k}$ values per se, but we took these effect sizes into account for inferring molecular mechanisms. If the number of mediator-associated instruments was sufficient ($\geq 3$) to conduct a univariable MR from the mediator to the outcome, we estimated $\alpha_{MY,k}$ from this analysis instead. In fact, the (marginal) contribution of an individual mediator can be better disentangled in univariable analyses, when mediators are highly correlated.

As our MVMR model assumes a chain of causal effects from the exposure to the mediator and then to the outcome, we conducted several Steiger filtering steps to reduce biases due to reverse causation. Although it has been proposed that DNAm could be a consequence of gene expression in the same locus[66], our model investigates the commonly assumed concept of DNAm regulating gene expression. In addition to meeting the Steiger criterion described above, exposure-associated IVs were required to pass that same threshold $t_{rev}$ of no larger mediator than exposure effects for each of the mediators $M_k$. Similarly, to mitigate reverse causal effects from the outcome on the mediators, mediator-associated instruments with larger $Y$ than $M$ effects were removed if not passing the $t_{rev}$ threshold. The SE of $\hat{\theta}_D$ was derived analogously to the univariable form (Eq. (2)) as shown in[19].

### MVMR sensitivity analyses
**Conditional F-statistic.** Conditional F-statistics of the exposure were calculated following the approach of Sanderson et al.[33]. This method involves the regression of the exposure on the mediators based on the IV effect sizes on each of these quantities. The residuals of this regression are then used to derive the conditional F-statistic. The original method additionally includes the phenotypic correlation matrix between the exposure and mediators, which we omitted by default due to the lack of these data and thus used the identity matrix instead. However, as a sensitivity analysis, we calculated conditional F-statistics incorporating the phenotypic correlations between transcript mediators. Transcript correlations were calculated on RNAseq data from the Cohorte Lausannoise (CoLaus) based on 555 samples[67]. Transcript correlations could be estimated for 19,517 transcript of which 15,021 overlapped with the eQTL dataset (Methods: Omics and trait summary statistics)[8]. We then calculated conditional F-statistics that included mediator correlations for all DNAm-trait pairs with at least 2 mediators and for which at least half of them had available correlation data. Conditional F-statistics >10 allow to reject the null hypothesis that the IVs are too weak to reliably estimate the multivariable effect of the exposure in the presence of the mediators.

**Heterogeneity Q-statistic.** Heterogeneity Q-statistics were computed as implemented in the TwoSampleMR package (v0.5.6, IVW-method)[34]. This test statistic quantifies the deviation of MR effect estimates of each individual IV from the IVW-estimate based on all IVs[68]. The null hypothesis of homogeneity within the IV set follows a chi-squared distribution with $m - 1$ degrees of freedom for the univariable MR, and $m - k$ degrees of freedom for the MVMR, where $m$ is the number of IVs and $k$ the number of mediators.

**Mediator selection threshold $P_{EM}$.** For transcripts to be included as mediators in the MVMR regression model they had to be i) in *cis* of the DNAm exposure probe ($\pm 500$kb) and ii) causally associated to the DNAm probe. This latter condition was verified by univariable MR analyses (Eq. (1)) of the DNAm exposure probe on each mediator transcript $k$ in the region estimating the effect sizes $\alpha_{EM,k}$ and p-values $P_{EM,k}$. Transcripts satisfying $P_{EM,k} < P_{EM}$ were included as mediators with the default threshold equalling 0.01. To assess the sensitivity of this threshold, we also tested milder and more stringent thresholds ($P_{EM}=0.05$ and 1e-3).

**Pleiotropy sensitivity analyses.** To quantify whether significant MR estimates between the exposure and mediators were observed due to horizontal pleiotropy, we conducted two sensitivity analyses. First, we repeated the mediation analysis excluding the top IV (i.e., exposure-associated IV with the lowest p-value) from both the total effect $\theta_T$ and direct effect $\theta_D$ calculations. This analysis allowed to assess whether mediation results are solely driven by a single strong IV. Second, we performed simulation analyses to quantify the possibility that causal links between DNAm probes and transcripts are driven by increased horizontal pleiotropy stemming from potential LD between methylation and transcript instruments due to their close genomic distance.

In the following, we outline step-by-step the workflow of the horizontal pleiotropy simulation study for which a schematic representation is shown in Supplementary Fig. 16. First, we considered DNAm-transcript pairs with a significant MR effect at $P_{EM} < $1e-6. For each of these selected DNAm-transcript pairs, we first fixed the SNP-DNAm and SNP-transcript effects as observed in the data. Then, using near-independent significant *cis*-eQTLs ($r^2 < 0.05$, $P < $1e-6) with observed marginal (univariable) effect sizes $\boldsymbol{\beta_M}$ (a vector of size $m_M$) and the corresponding pair-wise local LD matrix $\mathbf{C}_M$, we calculated the multivariable SNP effects on the transcript, $\boldsymbol{\beta}_{multi}$, as:

$$\boldsymbol{\beta}_{multi} = \mathbf{C}_M^{-1}\boldsymbol{\beta_M} \qquad (4)$$

Using the original data, we performed DNAm-transcript MR on $m_E$ exposure (i.e., DNAm-associated) IVs, yielding the causal effect $\alpha_{EM}$ with corresponding $p$-value, $P_{EM}$. We then performed simulation analyses as follows to obtain MR effects for a hypothetical transcript with identical multivariable eQTL effect size distribution as the real transcript. To achieve this, for each simulation $j$, we randomly selected $m_M$ leniently pruned ($r^2 < 0.5$) SNPs and assigned $\boldsymbol{\beta}_{multi}$ as their multivariable eQTL effects. Hence the marginal SNP-transcript effects for the $m_E$ exposure-associated SNPs can be calculated as follows:

$$\boldsymbol{\beta}_{marginal,j} = \mathbf{C}_{E,M_j}\boldsymbol{\beta}_{multi} \qquad (5)$$

where $\mathbf{C}_{E,M_j}$ is the LD-matrix between the $m_E$ exposure-associated SNPs and the $m_M$ randomly chosen SNPs (with multivariable SNP-transcript effect $\boldsymbol{\beta}_{multi}$). This way we assign marginal SNP-transcript effect sizes for the $m_E$ exposure-associated instruments, while keeping the multivariable eQTL effect size distribution identical to the one observed for the real transcript (but they are assigned to other SNPs). Univariable DNAm-transcript MR analyses could then be conducted (Eq. (1)) for each hypothetical transcript $j$, by using $\boldsymbol{\beta}_{marginal,j}$ as the outcome effect size vector. Thus, we generated MR estimates ($\alpha_{EM,j}$ and $P_{EM,j}$) for 100,000 ($N_{sim}$) hypothetical transcripts for 100 randomly selected DNAm-transcript pairs throughout the genome. The simulation p-value was then derived as $P_{sim} = \#(P_{EM,j} < P_{EM})/N_{sim}$.

**DNAm-to-trait mediation analysis**
A diagram of the workflow with each of the following steps is shown in Supplementary Fig. 1. First, univariable MRs were conducted to estimate the total causal effect $\hat{\theta}_T$ of the DNAm sites on each trait. We assessed the impact of ~50,000 DNAm probes with $\geq 5$ near-

independent ($r^2 < 0.05$) mQTLs after harmonization of the datasets. DNAm probes significantly associated to the outcome ($P_T < 0.05/50000=$1e-6) were clumped based on the p-value of the total causal effect $\hat{\theta}_T$, $P_T$ (distance-pruning at 1 Mb), to be independent of each other.

Second, MVMR analyses were performed to estimate the direct effect $\hat{\theta}_D$. Selected transcripts (see "Mediator selection threshold $P_{EM}$") were included as mediators as well as their associated SNPs as additional instruments. Steiger filtering on mediator-associated IVs was applied using the same $t_{rev}$ threshold as for exposure-associated IVs. Remaining IVs were then clumped based on a rank score determined as follows: 1) for each mediator, IVs were ranked according to their association p-value to the mediator and assigned an integer score, 2) for each IV, a final score was calculated as the sum of its individual mediator scores. Following the establishment of the $\boldsymbol{B}$ effect size matrix, $\hat{\theta}_D$ was calculated, as well as $\hat{\theta}_{D,top}$ which was estimated from a MVMR model that includes the transcript with the lowest $P_{EM,k}$ as sole mediator. If no transcript causally associated with the DNAm probe, mediation is not detectable, and hence $\hat{\theta}_D$ was set to $\hat{\theta}_T$ for that probe (inclusion of such probes in MP calculation was termed "overall mediation proportion"). As the Steiger filter removed exposure-associated instruments with larger mediator than exposure effects (see "Univariable and multivariable Mendelian randomization"), the number of initial exposure-associated instruments ($m_E \geq 5$) could decrease. Therefore, to avoid scenarios of reverse causality where the mediator exerts an effect on the outcome through the exposure, we required $\geq 3$ exposure-associated IVs.

We additionally conducted mediation analyses on independent mediators. To this end, selected mediators (those that passed $P_{EM}$) were clumped at various correlation thresholds $R_{med}$ (default $R_{med} < 0.3$, with 0.2 and 0.1 being tested as well). Correlations among mediators were calculated based on QTL effect sizes of independent exposure and mediator IVs and priority was given to the mediator with the lowest $P_{EM,k}$.

**Estimating and comparing mediation proportions**
Mediation proportions (MPs) were estimated on sets of DNAm-trait pairs with significant total causal effects $\hat{\theta}_T$, either grouped by trait (if there were at least 10 such pairs within a given trait), trait category (e.g. hepatic traits, inflammatory traits/diseases) or combining all pairs together. MPs were then calculated by regressing $\hat{\theta}_D$ on $\hat{\theta}_T$ (without intercept) to estimate for the unmediated proportion, $\hat{\gamma}$, which after correcting for regression dilution bias[31] (Eq. (6)):

$$\hat{\gamma}_{cor} = \frac{\hat{\gamma}}{\sqrt{1 - \frac{\sum SE^2(\hat{\theta}_T)}{\sum \hat{\theta}_T^2}}} \qquad (6)$$

yielded $\widehat{MP} = 1 - \hat{\gamma}_{cor}$ for a defined set of DNAm-trait pairs, together with a standard error. For individual DNAm-trait pairs, we report the $\widehat{MP}$ as $1 - \hat{\theta}_D/\hat{\theta}_T$, without providing its variance estimate since this would require individual-level data[26]. Note that $\widehat{MP}$ is an estimator of the true underlying MP and values outside the expected [0-1] range can be observed, especially if $\hat{\theta}_D$ and $\hat{\theta}_T$ estimates are of opposite sign. Such situations are expected to be rare in our analysis, as the total effect would be expected to be small and hence non-detectable.

In our approach, indirect effects $\theta_M$ are estimated by subtracting direct effects from total effects, which is also referred to as the difference in coefficients method[26]. Alternatively, the indirect effect can be estimated by the product of coefficients method[26], where univariable MR estimates from the exposure on the mediator are multiplied with the direct effects of the mediator on the outcome (Eq. (3)) and summed across mediators. Direct effects of the exposure on the outcome can then be obtained by the difference between the total and

indirect effect. As demonstrated earlier[26], the two approaches yield highly concordant results (Supplementary Fig. 20).

To test the statistical significance between $\widehat{MP}$s estimated on two different sets of exposure-trait pairs (e.g. $\widehat{MP}$ of a given physiological category vs all categories combined) or on the same exposure-trait pairs, but with different parameter settings (e.g. changing $P_{EM}$), we made use of $\hat{\gamma}$ and its corresponding standard error $\mathrm{se}(\hat{\gamma})$ obtained from regressing $\hat{\theta}_D$ on $\hat{\theta}_T$ (both of which being corrected for regression dilution bias (Eq. (6))) to yield $\hat{\gamma}_{cor}$ and $\mathrm{se}(\hat{\gamma})$. We then performed a two-sided z-test based on the following test statistic:

$$\frac{\hat{\gamma}_{cor}^{(1)} - \hat{\gamma}_{cor}^{(2)}}{\sqrt{\mathrm{se}\left(\hat{\gamma}_{cor}^{(1)}\right)^2 + \mathrm{se}\left(\hat{\gamma}_{cor}^{(2)}\right)^2}} \sim \mathcal{N}(0,1) \tag{7}$$

Significant difference between $\widehat{MP}$s was defined by a two-sided p-value ≤ 0.05. Of note, this z-test assumes independence between $\hat{\gamma}^{(1)}$ and $\hat{\gamma}^{(2)}$ which is not always guaranteed (i.e., when comparing $P_{EM}$ thresholds), hence the resulting p-values may be lenient.

### Omics and trait summary statistics

We used mQTL data from the GoDMC consortium ($n$=32,851)[6], which contains > 170,000 whole blood DNAm sites with at least one significant *cis*-mQTL ($P < $1e-6, < 1 Mb from the DNAm site, $n > 5000$). *Cis*-eQTL data were taken from the eQTLGen consortium ($n = 31,684$)[8] which includes *cis*-eQTLs (< 1 Mb from gene center, 2-cohort filter) for 19,250 transcripts (16,934 with at least one significant *cis*-eQTL at FDR < 0.05 corresponding to $P < $1.8e-05).

GWAS summary statistics for outcome traits came from the largest ($n_{average} > 320,000$), predominantly European-descent, publicly available studies, as listed in Supplementary Data 1. Thirty-seven out of the 50 traits were continuous biomarkers or continuous physical measures with the GWAS conducted on the UK Biobank[69] (http://www.nealelab.is/uk-biobank). Remaining GWAS data came mostly from case/control studies made available by the consortium of the respective disease. For binary outcome traits, log-odds ratios were used as effect sizes and results should be interpreted on the liability scale.

Prior to each mediation analysis, exposure and mediator omics, GWAS and the reference panel data were harmonized. The analysis was conducted on autosomal chromosomes, and palindromic single nucleotide variants (SNPs), as well as SNPs with an allele frequency difference > 0.05 between any pairs of datasets were removed. If allele frequencies were not reported by the GWAS summary statistics, allele frequencies from the UK Biobank were used. Z-scores of summary statistics (molecular and outcome GWAS) were standardized by the square root of the sample size to be on the same SD scale.

### DNAm-to-transcript MR analysis

As follow-up analyses, we calculated MR causal effects between all available DNAm sites and transcripts in *cis* (± 500 kb) following the same procedure as in the univariable MR to obtain total effects $\hat{\theta}_T$. First, near-independent ($r^2 < 0.05$) and significant ($P < $1e-6) exposure IVs were selected and IVs not passing the aforementioned Steiger filter were discarded. MR causal effects were then computed based on Eq. (1) for pairs with ≥3 exposure IVs.

Pearson correlation coefficient with previously reported DNAm-transcript correlations[37] was calculated on common DNAm-transcript pairs to explore agreement. DNAm probe annotations with respect to the assessed transcript were from the IlluminaHumanMethylation450kanno.ilmn12.hg19 R package (v0.6.1)[70].

### Simulation studies

We conducted simulation studies to assess the robustness of our model and to identify sources of bias in the estimated MP. Simulation settings were set up post-hoc to replicate mediation results obtained for real data (Supplementary Figs. 4-5; Supplementary Table 1).

We considered an exposure with heritability $h_E^2$ and $m_E$ independent IVs. Effect sizes $\beta_i^E$ for $m_E$ IVs were drawn from a normal distribution $\beta_i^E \sim \mathcal{N}(0, \sqrt{h_E^2/m_E})$ and rescaled to total $h_E^2$. $N_{med,pot}$ potential mediators were simulated, among which $N_{med}$ were contributing to the indirect effect $\theta_M$. Each mediator $k$ associated with $m_M$ IVs with direct effects $\beta_{direct,i}^{M_k} \sim \mathcal{N}(0, \sqrt{h_{M,direct,k}^2/m_M})$ rescaled to $h_{M,direct,k}^2$, the direct heritability of the mediator that does not take into account the additional heritability coming through the exposure. Causal effects of the exposure on the mediator ($\alpha_{EM,k}$) and of the mediator on the outcome ($\alpha_{MY,k}$) for $N_{med}$ mediators were drawn from a bivariate normal distribution $\alpha_{EM,k}, \alpha_{MY,k} \sim \mathcal{N}(\mathbf{0}, \boldsymbol{\Sigma})$ with $\boldsymbol{\Sigma}$ the covariance matrix:

$$\boldsymbol{\Sigma} = \begin{bmatrix} \mathrm{var}(\alpha_{EM}) & \rho \cdot \sqrt{\mathrm{var}(\alpha_{EM}) \cdot \mathrm{var}(\alpha_{MY})} \\ \rho \cdot \sqrt{\mathrm{var}(\alpha_{EM}) \cdot \mathrm{var}(\alpha_{MY})} & \mathrm{var}(\alpha_{MY}) \end{bmatrix}$$

where $\rho$ is the correlation between $\alpha_{EM,k}$ and $\alpha_{MY,k}$. For the remaining $N_{med,pot} \cdot N_{med}$ mediators, $\alpha_{EM,k}$ and $\alpha_{MY,k}$ causal effects were set to zero. The vector of effect sizes $\boldsymbol{\beta}^{M_k}$ of size $m_E + N_{med} \cdot m_M$ for each mediator $k$ was constructed to have effect sizes equalling $\beta_i^E \cdot \alpha_{EM,k}$ for $m_E$ exposure SNPs and effect sizes equalling $\beta_{direct,i}^{M_k}$ for $m_M$ mediator-associated SNPs. The effect sizes of remaining IVs associated to mediators $i \neq k$ were set to zero. Likewise, effect sizes of the $N_{med} \cdot m_M$ IVs on the exposure in the $\boldsymbol{\beta}^E$ vector were set to zero.

The indirect effect $\theta_M$, direct effect $\theta_D$ and total effect $\theta_T$ were calculated as:

$$\theta_M = \sum_k \alpha_{EM,k} \cdot \alpha_{MY,k} \; ; \; \theta_D = \theta_M \left(\frac{1}{MP} - 1\right) \; ; \; \theta_T = \theta_D + \theta_M$$

These quantities allowed to generate the outcome effect size vector $\boldsymbol{\beta}^Y$:

$$\boldsymbol{\beta}^Y = \theta_D \cdot \boldsymbol{\beta}^E + \sum_k \alpha_{MY,k} \cdot \boldsymbol{\beta}^{M_k}$$

For each scenario, we simulated 500 data sets to each time get $\boldsymbol{\beta}^E$, $\boldsymbol{\beta}^{M_k}$ and $\boldsymbol{\beta}^Y$. Normally distributed noise, as a function of the sample size $N$, $\epsilon_i^E \sim \mathcal{N}(0, 1/N_E)$, $\epsilon_i^M \sim \mathcal{N}(0, 1/N_M)$ and $\epsilon_i^Y \sim \mathcal{N}(0, 1/N_Y)$ was added to each simulated vector. To approximate our real data, exposure effect sizes of SNPs serving as mediator instruments were set to zero again. We then estimated for each model $\hat{\theta}_T$ and $\hat{\theta}_D$ by including mediators that satisfied $P_{EM}$ (p-value of the causal effect from the exposure on the mediator) denoted $N_{med,sig}$. Causal effects $\hat{\theta}_D$ were regressed on $\hat{\theta}_T$ to estimate the coefficient $\hat{\gamma}$ which after accounting for regression dilution (Eq. (6)) allowed to obtain the estimated $\widehat{MP}$.

### Reporting summary

Further information on research design is available in the Nature Portfolio Reporting Summary linked to this article.

## Data availability

Methylation QTLs used in this study are from the GoDMC mQTL meta-analysis and are available on the GoDMC Consortium website (http://mqtldb.godmc.org.uk/downloads). Expression QTLs are from the eQTLGen eQTL meta-analysis and are available on the eQTLGen Consortium website (https://www.eqtlgen.org/cis-eqtls.html). The list of GWAS summary statistics used in this study is in Supplementary Data 1, all of which are publicly available. UK10K individual-level data are available upon request (https://www.uk10k.org/data_access.html). Source data are provided with this paper.

## Code availability

Software to conduct univariable MR-IVW (molecular trait → outcome, molecular trait 1 → molecular trait 2) and multivariable MR-IVW (molecular trait 1 → molecular trait 2 → outcome) is available at https://github.com/masadler/smrivw(https://doi.org/10.5281/zenodo.7324709[71]). Source code (C++, released under GPL v2 license) and executable file (for Linux platforms, released under MIT license) are provided which rely on functionalities and the data management architecture of the SMR software v1.03 (https://cnsgenomics.com/software/smr[24]). The provided documentation hosted on the GitHub repository guides users in reproducing the mediation results and conducting univariable and multivariable MR on their own combinations of QTL and GWAS datasets.

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

## Acknowledgements

This work was supported by the Swiss National Science Foundation (310030_189147) to Z.K. L.D. was calculated based on the UK10K data resource (EGAD00001000740, EGAD00001000741). Computations we performed on the JURA cluster of the University of Lausanne. We have used RNA-seq data from 557 CoLaus participants to compute gene-gene correlations, which were kindly made available by Sven Bergmann.

## Author contributions

M.C.S., E.P. and Z.K. conceived and designed the study. M.C.S. performed statistical analyses. K.L. contributed to the statistical analyses. EP provided guidance on statistical analyses. Z.K. supervised all statistical analyses. All the authors contributed by providing advice on interpretation of results. C.A. contributed with the biological interpretation of the results. M.C.S., E.P. and Z.K. drafted the manuscript. C.A. contributed to the writing of specific sections. All authors read, approved, and provided feedback on the final manuscript.

## Competing interests

The authors declare no competing interests.
