## [Peer Review File · Nature Communications]

Quantifying the role of transcript levels in mediating DNA methylation effects on complex traits and diseasesREVIEWER COMMENTS

Reviewer #1 (Remarks to the Author):

The authors propose a pipeline using multivariable MR to identify DNA methylation sites with effects on complex traits that are mediated by gene expression. I think the approach is thoughtful and reasonable and the question is interesting.

A weakness of the approach is that it relies on multiple ad-hoc thresholding steps and lacks clarity about how to choose these thresholds and under what conditions we can expect accurate results.

Major comments:

1. The method contains many thresholding and filtering steps so it is hard to provide theoretical guarantees. It would be nice to have more discussion of how to pick the thresholds described and under what conditions the method is more or less biased. In the absence of theoretical results, the reliability of the method really depends on how accurately the simulations mimic the real data. - I would suggest either providing more theoretical justification of the ad-hoc elements of the procedure or spending more time justifying parameter choices in the simulations and exploring a wider range of simulations (or both). The simulations have six exposure variants that are expected to explain 35% of the variation of the methylation trait. Is this realistic based on the data that you saw? It is very large for complex traits but I would guess that methylation traits have a smaller number of large effect variants. It may be useful to vary the relative strength of exposure and mediator variants relative to each other (more on this in Q2).

2. Another major concern is weak instrument bias. In particular, you may have more bias in the MVMR step than in the univariable MR step. Can you show in simulations how the relationship between the F-statistic in the univariable stage and the conditional F-statistic in the multivariable stage relate to bias in the MP estimate? I was glad to see a discussion of this issue in the real data (supp Fig 8). It would be good to augment this with simulation results so that you can identify which estimates you expect to be the most biased.

3. I think the other major issue in my mind is pleiotropy. With a small number of IVs, heterogeneity tests may fail to detect outliers. Methylation probes and transcripts are spatially close together. This means it would not be too surprising to see an mQTL very near an eQTL leading to many SNPs associated with both methylation and expression simply due to LD. Two questions related to this:
3a. In figure 5, it looks like signals may be driven by only one variant and the remaining variants could be quite weak. Is this the case? Do those signals persist without the top variant? How many methylation-transcript-trait connections are driven by a single SNP?
3b. You might be able to assess the expected amount of transcript association expected for a random SNP in a region by doing something similar to a permutation test. Pick a methylation probe-transcript pair for which you have identified a methylation \rightarrow expression effect. Suppose you have n_M IVs (mQTLs) for the methylation probe with effect estimates $\hat{\beta}_M$ and transcript effect estimates $\hat{\beta}_T$. Randomly sample N_m variants that are not mQTLs but which are a similar distance from the probe as the actual mQTLs. This new set of variants will have transcript associations $\hat{\beta}_{T,j}$ where j indexes the re-sampling procedure. You can compare $\hat{\beta}_{T,j}$ to $\hat{\beta}_T$ and you can compute the methylation \rightarrow transcript effect estimate you would get if you replaced $\hat{\beta}_T$ in the MR procedure with $\hat{\beta}_{T,j}$. Hopefully this will show that random SNPs are less associated with the transcript than the mQTLs are demonstrating that not all of your effect is due to pleiotropy.

4. I think I may be missing something about Figure 5. In Figure 5a, the mediated effect appears to be $-0.67 \times -0.13 = 0.087$. According to Figure 1 and the description, $MP = \theta_m / \theta_t = 0.087 / 0.11 = 0.79$. But the figure lists MP as 0.47. Is the MP given in the figure referring to something else? Is it

an average over multiple methylation probes rather than only the probe shown in the figure? Same question about 5b in which it appears MP should be equal to 0.33.

5a. I was generally confused about when the estimated MP was an average and when it was a ratio. My understanding is that in Figure 3, the barplots are showing the average MP over all methylation probes with detectable mediation for a particular trait. Is there a reason to believe that all methylation probes should have the same mediated proportion? This MP isn't clearly described as an average, giving the impression that it is measuring a specific causal parameter. How much variability in the ratio MP estimates did you find within traits?

5b. On the same theme, I think it would be worth clarifying that the 37.8% reported in the abstract is an average only over DNAm probes with any detectable mediation. This is very different from saying that 37.8% of methylation effects on the traits are mediated by transcripts.

6. Should eq (5) be a t-distribution and not a standard normal? This equation contains a typo, I think one of the terms in the denominator should be $\hat{\gamma}^2$.

7. In equation 4, if you are estimating the MP for a single methylation probe-trait pair, is it correct that $\hat{\gamma}$ is just the ratio θ_D/θ_T ? Do you never end up computing the MP for only a single probe (back to the confusion of question 5)? If you did compute this value, $\hat{\gamma}$ would no longer be approximately normal and would not have moments as it is the ratio of two normals. It may be that you never encounter this problem because you are always averaging over many probes. That point is unclear to me.

Reviewer #2 (Remarks to the Author):

Thank you for the opportunity to review this interesting manuscript applying multivariable Mendelian randomisation to quantify the role of transcript levels in mediating DNA methylation on outcomes. The manuscript has a number of strengths including comprehensive analyses and clarity. It also offers an example of approaches that can be used to better understand the pathway from genotype to disease via multiple layers of intermediate molecular traits. Below, I list a number of questions that I have regarding the manuscript.

How the instruments are selected could be made clearer. When I first read the manuscript, I thought that instruments for the exposure (DNAm) were selected rather than instruments for both the exposure and the mediator (transcript levels). Reading the method section, I believe that the latter is true, which makes more sense, maybe this can be made explicit earlier on, for example near figure 1.

In classical mediation analyses, many researchers tend to preselect significant relationships to be modelled in the mediation framework. The selection is either transparent or, what is worse, does not appear in the final article. This is obviously an issue: if you compute a 100 correlations, 5 of them will be significant by chance, and any mediation model applied to these variables is likely to yield significant mediation effects were none are present. I know the authors are aware of this issue, but I think their approach could be reported more clearly. For example, they pre-filter DNAm probes that are causally related to the outcome, and transcripts that are causally related to DNAm.

“MP estimates were then computed only for DNAm-trait pairs with significant Bonferroni-corrected θ effects, grouped by trait, trait category and all pairs combined. We further filtered out DNAm-trait pairs whose exposures have no significant causal effect on any potential transcript as the link”

The selection is drastic, e.g. around 2000 DNAm causally related to outcomes out of 1 million DNAm to outcomes relationships if I got this properly. This should naturally lead to more significant mediation findings in this subset and likely exaggerate the MP (mediated proportion)? The authors propose a number of sensitivity analyses, which may already tackle this issue. However I would appreciate:

1. A clear reporting in one place in the manuscript of all numbers involved at the different stages of filtering (i.e. initial number of probes; number after filtering for causal relationship with the outcome; initial number of transcripts etc...). I think most of these numbers are already present in the manuscript but in different places. Maybe a comprehensive flowchart would be useful.

2. A more thorough discussion of how filtering at different stages might impact the results, not only for MPs but also potentially false positives. The authors have done multiple testing correction at different stages, but does that fully account for the problems mentioned above (i.e. only carrying forward significant bivariate relationships in the mediation model).

A related comment is on the accuracy, or even the relevance of the MPs that are reported. As the authors mention, MPs are very dependent on technical issues. For example, MPs are higher when the transcript is closer to the DNAm site. But transcripts are imperfectly measured (i.e. as the authors mention available data is a magnitude lower than the transcriptome). Can we imagine that if we had the perfect dataset, MPs would be close to 1 all the time? In brief, there is a possibility that the authors choices (selection, see above) might overestimate MPs, but that the limitations of the datasets, might underestimate MPs. If this is the case, how reliable and meaningful are the MPs results? I wonder whether the authors need to insist so much on MPs, which are currently an important focus of the manuscript. Their approach seems more interesting to me to discover pathways (of which the authors give a few examples) rather than to estimate abstract quantities like MPs? Also, the authors selected only probes with significant effects on outcomes. However, a mediated effect can be observed even in the absence of a univariate effect because of suppression effects (e.g., two indirect paths via transcript levels in opposite direction and summing up to 0). The authors mentioned that the relationships between DNA methylation and transcript levels was not always negative as expected, in how many cases did they have different signs in the DNAm -> transcript paths for a give probe (e.g. 3 transcript levels, 1 related positively to DNAm and 2 negatively) and does this affect their findings.

The authors look at reverse causation by implementing Steiger filtering for both the DNAm-> outcome and DNAm -> transcript, which is important. But their theoretical model is still DNAm -> transcripts -> outcome rather than transcript -> DNAm -> outcome. Maybe it could be made more explicit for the reader less familiar with basic biology, why this is the chosen model.

If we have three paths in the multivariate model: a (DNAm -> transcript) b (transcript -> outcome) and c' (direct effect), with c being the univariate effect (DNAm -> outcome). In classical mediation analysis, there are two ways of computing the proportion mediated: (i) as the authors did $(1-c')/c$ or (ii) $a*b/c$. Can the authors say: (i) is it possible to use the second approach, which is usually preferred in classical mediation analysis, within MVMR; (ii) why did they choose the first approach over the second. Related to that is the question of whether estimates actually sum up to 1? I.e. given that you have different instruments for the exposure and the mediator, if you were to compute a and b, would $a*b + c' = c$.

Binary outcomes like schizophrenia are mentioned but I have seen no mention of whether there are specific considerations to take into account when using binary outcomes (i.e. the outcome gwas would include log odds).

Can the authors mention more specifically, for example in the methods section of the discussion, if there are any novel statistical aspects in their methods, or if it is applying existing MVMR methods.

The authors computed heterogeneity statistics to test for pleiotropy. Is there more sophisticated way to do this. I assume that MR Egger would not work because of the number of instruments? But any other methods, for example to remove outliers, like MR-PRESSO?

The authors say in the discussion

"Their results indicated that colocalization between DNAm sites and complex traits were typically due to horizontal pleiotropy as opposed to DNAm mediating the GWAS effect. These results confirm that colocalization analysis is not necessarily able to detect causal relationships"

I'm not sure I understand here "DNAm mediating the GWAS effect". This doesn't appear to be the same question as it's about whether DNAm mediates SNP effects on the outcome rather than whether transcripts mediate DNAm effects on outcome. How is that reference relevant to the current work.

Also, isn't that the case that colocalization is better at identifying relationships that are due to LD rather than true biological pleiotropy (i.e. same variant at play), which MR doesn't really do? I think it would be important to include a lengthier and more nuanced discussion of the respective limitations and strength of the different methods that the authors mention. This will allow the authors to highlight the interest of their approach and how it complements existing ones.

"The original method additionally includes the phenotypic correlation matrix between the mediators, however, due to the lack of these data, we used the identity matrix instead."

The identity matrix would assume a correlation of zero between mediators? Isn't that unlikely? The simple fact that different transcripts are influenced by the same DNAm site would generate a non-null phenotypic relationship between them. There must be accessible data providing a general sense of how correlated expression data are, which would provide a better approximation?. Or maybe a sensitivity analysis using a different correlation matrix.

Minor:

In Figure 1, arrows could be clearer. For example, the θ_M should have an arrow leading to transcription and another arrow from transcription to outcome, consistent with the classical way of representing mediation.

The authors say:

"Surprisingly, 46.6% of significant DNAm-to-transcript effects were 233 of positive sign (i.e. DNAm increasing transcription), particularly so when the DNAm site was situated in the gene body (enrichment p-value = $2.15e-10$). This observation is in line with previous genome-wide methylation and gene expression association studies which reported high fractions of positive correlations (30-35%)^{36,3}"

How is it surprising if in line with prior investigations? Maybe change the phrasing.

Four Steiger filtering, the authors mention "trev set at -2" I don't understand why it is 'minus' 2. The numerator is a difference of absolute values, where the first should be higher (i.e. relationship with expression) than the second (relationship with outcomes), hence this quantity should be positive? If it is negative, it means that the relationship with the outcome is superior, which is a problem. I am probably missing something, but maybe clarifying the text would be useful.

"If the number of mediator-associated instruments was sufficient (≥ 3) to conduct a univariable MR from the mediator on the outcome, we estimated $\alpha_{MY,k}$ from this analysis instead, since computed on a single regressor, narrower CIs are obtained."

I don't understand how can the univariate MR can provide the same estimate as the multivariate MR as the effect of mediator on outcome should be adjusted for the exposure in the latter?

Color code:

Answers to the reviewers are written in green.

Unchanged elements borrowed from the manuscript are in blue (italic), and changed text quoted from the manuscript is in brown (italic).

Reviewer #1

The authors propose a pipeline using multivariable MR to identify DNA methylation sites with effects on complex traits that are mediated by gene expression. I think the approach is thoughtful and reasonable and the question is interesting.

A weakness of the approach is that it relies on multiple ad-hoc thresholding steps and lacks clarity about how to choose these thresholds and under what conditions we can expect accurate results.

Major comments:

1. The method contains many thresholding and filtering steps so it is hard to provide theoretical guarantees. It would be nice to have more discussion of how to pick the thresholds described and under what conditions the method is more or less biased. In the absence of theoretical results, the reliability of the method really depends on how accurately the simulations mimic the real data.

- I would suggest either providing more theoretical justification of the ad-hoc elements of the procedure or spending more time justifying parameter choices in the simulations and exploring a wider range of simulations (or both). The simulations have six exposure variants that are expected to explain 35% of the variation of the methylation trait. Is this realistic based on the data that you saw? It is very large for complex traits but I would guess that methylation traits have a smaller number of large effect variants. It may be useful to vary the relative strength of exposure and mediator variants relative to each other (more on this in Q2).

We revised our simulations to better reflect real data settings and dedicated more space to the simulation analyses to cover a wider range of parameter values. For each of the simulation parameters (e.g. number of instrumental variables, exposure and mediator heritabilities, number of relevant mediators, etc.), we show their distribution as observed in real data (i.e., DNAm → trait mediation analysis through at least 1 transcript in cis). The distributions are shown in a new Supplementary Figure 5 and numerical summaries of the interquartile ranges and means are shown in a new Supplementary Table 2 (please see below).

We extended the simulation studies to cover at least the interquartile range when varying the respective parameter while keeping other parameters constant at the median value observed in real data applications (Supplementary Table 3). Furthermore, we put a stronger emphasis on the bias introduced by weak instruments. We varied the exposure and mediator heritabilities (i.e., their respective strength) in the simulation studies and showed how this affected the estimated mediation proportion. In this context, we now also compute conditional F-statistics in our simulation analyses and show how it can help to identify weak instrument bias in case of low (conditional) mediator heritabilities.

As mentioned by the reviewer, it may seem that six exposure variants explaining 35% of the DNAm heritability is very large. Supplementary Figure 5c now clearly shows that the DNAm exposures in the mediation analyses have high heritabilities (median of 32%) and are not very

polygenic (median of 4 independent exposure variants). However, we included simulation studies where we varied the polygenicity at different exposure heritabilities to cover more extreme scenarios (Supplementary Figure 7). In these analyses, we observed that low heritability results in the underestimation of the mediation proportions (\widehat{MP}) whereas the polygenicity has no discernible impact (possibly due to the fact that while higher polygenicity reduces the univariate F-statistic, it increases the number of included instruments which may improve the conditional F-statistic).

We updated Figure 2 to show these additional analyses, and the result section about the simulation studies (“Simulation results”) in the manuscript reads now as follows:

We performed simulation studies to assess the bias in estimated MPs (\widehat{MP}) by exploring a wide range of realistic parameter settings which cover at least the interquartile range as observed in real data (Supplementary Figs. 4-5; Supplementary Tables 2-3; Methods). Using default settings (i.e., median values for each parameter such as 2 true mediators N_{med} and a true MP of 35%), the bias in \widehat{MP} is minimal with the mean \widehat{MP} equalling 33.5% (95% CI: [32.0%-35.0%]; Supplementary Fig. 6, Table 2).

A determining factor in accurately estimating MPs was the available sample size to derive the mediator QTL effects. Low sample sizes resulted in significant underestimations of the MP, with mediator sample size of 3,000 compared to 30,000 resulting in a 17% relative decrease (6% in absolute values) of the estimated \widehat{MP} (Fig. 2a). The reason for this significant underestimation was not only weak instrument bias, but also the omission of relevant mediators with on average only 1.17 ($N_{med,sig}$) out of the 2 (N_{med}) relevant mediators detected at a sample size of 3,000 (Fig. 2a). We further tested the robustness of the \widehat{MP} with respect to the number of included mediators by varying the mediator selection threshold P_{EM} . Among a set of 20 potential mediators, those not passing the P_{EM} as determined by univariable MR effects of the exposure on each of these mediators were excluded from the MVMR model (Methods). Increasing P_{EM} resulted in more, and potentially irrelevant mediators to be included into the MVMR regression, which did not bias the \widehat{MP} (Fig. 2b). However, at too lenient p-values, \widehat{MP} s started to become underestimated (due to the inclusion of too many non-mediators in the model) as well as at too stringent p-values, a consequence of missing relevant mediators (Fig. 2b). The used mQTL and eQTL datasets provide SNP effect sizes in cis of the assessed DNAm probe and transcript levels, respectively, and were primarily restricted to significant mQTLs for the former. Thus, in the MVMR analysis SNP-exposure effects for mediator instruments are often non-significant (hence unreported) and set to zero to reduce regression dilution bias (i.e. weak instrument bias). Our simulation studies, which mimicked this scenario by setting non-significant effects to zero (Methods), confirmed that this did not introduce any bias.

Furthermore, we investigated weak instrument bias of both exposure- and mediator-associated IVs. When mediator-associated IVs were weak (i.e., low direct mediator heritabilities ($h_{M,direct}^2$; Methods), a high variability and significant underestimation of the \widehat{MP} was observed (Fig. 2c). In case of low mediator heritability, the conditional F-statistic of the exposure was also below the critical threshold of < 10 (Methods) indicating weak instruments. Similarly, for low exposure heritability (h_E^2), underestimated \widehat{MP} s were obtained, even in case of high conditional F-statistic (> 120; Fig. 2d). Additional simulation studies with more polygenic exposures and increased number of relevant mediators N_{med} for different exposure and mediator heritabilities corroborated the findings of underestimated \widehat{MP} s in case of weak instruments (Supplementary Fig. 7).

Figure 2. Simulation results to assess the bias in estimated MPs (\widehat{MP}) in real data settings. a Influence of the mediator sample size on the estimated \widehat{MP} (orange) and number of selected mediators ($N_{med,sig}$, blue). **b** Influence of the mediator selection threshold P_{EM} . **c, d** Sensitivity of \widehat{MP} s in settings of weak instruments, simulated by low mediator ($h^2_{M,direct}$) and exposure (h^2_E) cis-heritabilities, respectively. Conditional F-statistics of the exposure allow to test for weak instruments bias (critical values are defined at a threshold of F-statistic < 10). For a given parameter setting, 500 exposure-outcome pairs were simulated on which an \widehat{MP} and 95% CI (error bars) were estimated. The true MP of the model was 0.35 (horizontal orange lines), and the true number of relevant mediators N_{med} was 2 (horizontal blue lines) among a set of 12 potential mediators (20 in **b**).

Supplementary Figure 5. Distribution of the simulation parameters as observed in real data.

Numerical values (interquartile ranges, mean) are shown in Supplementary Table 2 and parameter choices of the different simulation settings to explore the full range of realistic parameter estimates are summarized in Supplementary Table 3. **a** Distribution of the number of exposure-associated (m_E) and mediator-associated (m_M) independent instrumental variables (IVs). **b** Distribution of the number of selected mediators ($N_{med,sig}$) and of the total number of potential mediators in the region ($N_{med,pot}$ with $N_{med,pot} \geq N_{med,sig}$). **c** Distribution of the exposure heritability (h_E^2) in relationship with the number of exposure-associated IVs (m_E). **d** Distribution of the direct heritability of each mediator k ($h_{M,direct,k}^2$) - ignoring the heritability coming through the exposure - in relationship with the number of mediator-associated IVs (m_M). **e** Distribution of the variance (across all mediators) of the exposure-to-mediator causal effects ($var(\alpha_{EM,k})$) and mediator-to-outcome effects ($var(\alpha_{MY,k})$) as estimated by $\hat{\alpha}_k^2 - se(\hat{\alpha}_k)^2$. **f** Distribution of the correlation (ρ) between $\alpha_{EM,k}$ and $\alpha_{MY,k}$. Estimation was done by considering DNAm-trait pairs with at least 3 mediators and calculating for each pair the correlation between $\alpha_{EM,k}$ and $\alpha_{MY,k}$ that were estimated for each mediator.

Supplementary Figure 6. Simulation results with the parameter default settings as indicated in Supplementary Table 2. 500 exposure-outcome pairs were simulated and for each a direct and total effect was estimated. The estimated mediation proportion (%) together with the 95% CI are displayed in the plot area with the slope plotted in black (blue line represents the identity line). Mediators were selected based on a p-value threshold P_{EM} and the distribution of the selected number of mediators (among a set of 12 potential mediators) is shown in the histogram. The true number of relevant mediators was 2 and the true MP was 35% (Supplementary Table 2).

Supplementary Figure 7. Simulation results varying m_E and N_{med} (Supplementary Table 3). **a** The number of exposure-associated instrumental variables (IVs) m_E was changed for different exposure heritabilities h_E^2 . The estimated mediation proportions were more dependent on h_E^2 than on the polygenicity of the exposure. **b** Dependence of the estimated mediation proportion on the number of true mediators N_{med} (i.e., mediators contributing to the indirect effect) stratified by $h_{M,direct}^2$. Underestimations were observed for fewer mediators (1-2) and when the direct mediator heritability was low (first quartile). With fewer true mediators N_{med} , missing a relevant mediator has a greater impact on the estimated mediation proportion than if multiple N_{med} are contributing towards the mediated effect.

Supplementary Table 2. Means and interquartile ranges of the simulation parameters as observed in real data. The full distribution of each parameter is shown in Supplementary Fig. 5.

	Quartile 1	Median	Mean	Quartile 3
$N_{med,pot}$	7	12	14.7	21
$N_{med,sig}$	1	2	3.3	4
m_E	3	4	5.09	6

m_M	1	3	5.65	8
h^2_E	0.179	0.319	0.403	0.539
$h^2_{M,direct}$	4.75E-03	0.0148	0.0418	0.047
$var(\alpha_{EM})$	5.48E-03	0.0196	0.0789	0.0751
$var(\alpha_{MY})$	1.18E-04	7.37E-04	9.52E-03	3.78E-03
ρ	-0.39	-0.0216	-0.0112	0.361

Supplementary Table 3. Values used in the different simulation settings to mimic mediation of DNAm-to-trait effects through transcript levels. Results of the default model are shown in Supplementary Fig. 6, results of varying the sample size N_M , the mediator selection threshold P_{EM} , and heritabilities $h^2_{M,direct}$ and h^2_E in Fig. 2, and the remaining simulation settings in Supplementary Fig. 7. Median parameter values are used in the default model and values comprising the interquartile range when varying the respective parameter.

	Default model (median values)	Varying N_M	Varying P_{EM}	Varying $h^2_{M,direct}$	Varying h^2_E	Varying m_E	Varying $N_{med,sig}$
N_{med}	12		20		12		20
$N_{med,sig}$			2				[1 - 10]
m_E			4			[3 - 12]	4
m_M			3				
h^2_E	0.319				[0.05 - 1]	0.1, 0.3, 0.5	0.319
$h^2_{M,direct}$	0.0148			[3E-04 - 0.64]	0.0148		4.75E-03, 0.0148, 0.047
ρ			-0.02				
P_{EM}	0.01		[1E-06 - 1]		0.01		
N_M	30,000	[100-100,000]			30,000		
N_E					30,000		
N_Y					300,000		
$var(\alpha_{EM})$					0.02		
$var(\alpha_{MY})$					1.00E-03		
MP					0.35		

2. Another major concern is weak instrument bias. In particular, you may have more bias in the MVMR step than in the univariable MR step. Can you show in simulations how the relationship between the F-statistic in the univariable stage and the conditional F-statistic in the multivariable stage relate to bias in the MP estimate? I was glad to see a discussion of this issue in the real data (supp Fig 8). It would be good to augment this with simulation results so that you can identify which estimates you expect to be the most biased.

Please see our answer to Q1, where we give a detailed account of the impact of all relevant parameters on the conditional F-statistic and report the resulting bias. Note however, that low conditional F-statistic is not the only source of bias, since missing potential mediators (due to lack of power) can lead to a more important drop in MP estimation.

In our real data application, we estimated the F-statistic in the univariable stage and conditional F-statistic in the multivariable stage. There was no significant correlation between these two quantities ($R = -0.037$, $P = 0.094$) when considering the 2,069 DNAm-trait pairs assessed for mediation. This is due to the substantial change of included instruments between the calculation of the uni- vs multivariable regression models. Furthermore, the minimum F-statistic in the univariable stage was 21.7 suggesting that weak exposure instrument bias is not an issue in the univariable MR analyses. Given these two observations and the fact that the instrumental variables can be very different in the univariable and multivariable MR regressions, we chose to limit the results and discussion to the conditional F-statistics in the multivariable stage.

3. I think the other major issue in my mind is pleiotropy. With a small number of IVs, heterogeneity tests may fail to detect outliers. Methylation probes and transcripts are spatially close together. This means it would not be too surprising to see an mQTL very near an eQTL leading to many SNPs associated with both methylation and expression simply due to LD. Two questions related to this:

3a. In figure 5, it looks like signals may be driven by only one variant and the remaining variants could be quite weak. Is this the case? Do those signals persist without the top variant? How many methylation-transcript-trait connections are driven by a single SNP?

3b. You might be able to assess the expected amount of transcript association expected for a random SNP in a region by doing something similar to a permutation test. Pick a methylation probe-transcript pair for which you have identified a methylation \rightarrow expression effect. Suppose you have n_M IVs (mQTLs) for the methylation probe with effect estimates $\hat{\beta}_M$ and transcript effect estimates $\hat{\beta}_T$. Randomly sample N_m variants that are not mQTLs but which are a similar distance from the probe as the actual mQTLs. This new set of variants will have transcript associations $\hat{\beta}_{T,j}$ where j indexes the re-sampling procedure. You can compare $\hat{\beta}_{T,j}$ to $\hat{\beta}_T$ and you can compute the methylation \rightarrow transcript effect estimate you would get if you replaced $\hat{\beta}_T$ in the MR procedure with $\hat{\beta}_{T,j}$. Hopefully this will show that random SNPs are less associated with the transcript than the mQTLs are demonstrating that not all of your effect is due to pleiotropy.

We thank the reviewer for the suggestions outlined above. In our revised manuscript, we extended the section "MVMR sensitivity analyses" to include these two sensitivity analyses assessing the possibility of MR associations arising from horizontal pleiotropy. Please find

below the additions we made in the manuscript and specific answers underneath each sub question.

We added a method subsection called “Pleiotropy sensitivity analyses” in the “MVMR sensitivity analyses” section detailing the two analyses:

To quantify whether significant MR estimates between the exposure and mediators were observed due to horizontal pleiotropy, we conducted two sensitivity analyses. First, we repeated the mediation analysis excluding the top IV (i.e., exposure-associated IV with the lowest p-value) from both the total effect θ_T and direct effect θ_D calculations. This analysis allowed to assess whether mediation results are solely driven by a single strong IV. Second, we performed simulation analyses to quantify the possibility that causal links between DNAm probes and transcripts are driven by increased horizontal pleiotropy stemming from potential LD between methylation and transcript instruments due their close genomic distance.

To address the second concern, we considered DNAm-transcript pairs with a significant MR effect at $P_{EM} < 1e-6$). For each of these selected DNAm-transcript pairs, we first fixed the SNP-DNAm and SNP-transcript effects as observed in the data. Using near-independent significant cis-eQTLs ($r^2 < 0.05$, $P < 1e-6$) with observed marginal (univariable) effect sizes β_M (a vector of size m_M) and the corresponding pair-wise local LD matrix C_M , we calculated the multivariable SNP effects on the transcript, β_{multi} , as:

$$\beta_{multi} = C_M^{-1} \beta_M \quad (4)$$

Using the original data, we performed DNAm-transcript MR on m_E exposure (i.e., DNAm-associated) IVs, yielding the causal effect α_{EM} with corresponding p-value, P_{EM} . We then performed simulation analyses to obtain MR effects for a hypothetical transcript with identical multivariable eQTL effect size distribution as the real transcript. To achieve this, for each simulation j , we randomly selected m_M leniently pruned ($r^2 < 0.5$) SNPs and assigned β_{multi} as their multivariable eQTL effects. Hence the marginal SNP-transcript effects for the m_E exposure-associated SNPs can be calculated as follows:

$$\beta_{marginal,j} = C_{E,M_j} \beta_{multi} \quad (5)$$

where C_{E,M_j} is the LD-matrix between the m_E exposure-associated SNPs and the m_M randomly chosen SNPs (with multivariable SNP-transcript effect β_{multi}). This way we assign marginal SNP-transcript effect sizes for the m_E exposure-associated instruments, while keeping the multivariable eQTL effect size distribution identical to the one observed for the real transcript (but they are assigned to other SNPs). Univariable DNAm-transcript MR analyses could then be conducted (Eq. 1) for each hypothetical transcript j , by using $\beta_{marginal,j}$ as the outcome effect size vector. Thus, we generated MR estimates ($\alpha_{EM,j}$ and $P_{EM,j}$) for 100,000 (N_{sim}) hypothetical transcripts for 100 randomly selected DNAm-transcript pairs throughout the genome. The simulation p-value was then derived as $P_{sim} = \#(P_{EM,j} < P_{EM}) / N_{sim}$.

The results of these two analyses have been added in the “MVMR sensitivity analyses” section which now accommodates the following paragraphs:

Finally, we conducted sensitivity analyses to determine whether significant MR associations were due to horizontal pleiotropy. Regulatory pathways between DNAm exposure probes and transcript mediators were assessed in cis. As such, SNPs in LD with significant QTLs for both quantities could give rise to an association merely because of horizontal pleiotropy (i.e., due to random overlap between cis-QTLs in close vicinity), an issue further exacerbated by the fact that molecular omics entities generally have fewer associated IVs than complex traits. To assess whether mediation results are only based on a single strong genetic instrument, we repeated the mediation analysis excluding the top IV (i.e., exposure-associated IV with the lowest p-value) from both the total effect θ_T and direct effect θ_D calculations (Methods). The results show that while MR effect estimates remain concordant in magnitude and effect direction, the estimates are noisier due to the much weaker instruments (significantly lower F-statistics; two-sided t-test: $P = 5.37e-11$; Supplementary Fig. 15). MP estimates were also somewhat higher when the top IV was excluded ($\widehat{MP} = 47.3\%$ (95% CI: [38.4%-56.2%]); $P_{diff} = 0.023$; Supplementary Fig. 15), however, this was no longer the case when controlling for conditional F-statistics > 10 ($\widehat{MP} = 40.9\%$ (95% CI: [29.3%-52.4%]); $P_{diff} = 0.48$). Additionally, we performed simulation analyses to assess the possibility of significant DNAm-transcript associations caused by cis-mQTL and -eQTL signals being in LD (Methods). The analysis shows that randomly picked eQTL-SNPs in the region result in slightly inflated, but much weaker MR associations than using the original eQTL data (Supplementary Fig. 16). The results indicate that by-chance LD between cis-mQTLs and -eQTLs can yield false positive findings, but those signals are substantially weaker than the ones observed in real data. In other words, mQTL and eQTL IVs are in much higher LD than expected by chance.

Overall, these sensitivity analyses showed that the estimated MPs remain robust when removing DNAm-trait pairs that potentially violate MVMR assumptions, while also suggesting that the set P_{EM} threshold of 0.01 may lead to underestimated MP estimates. Finally, we found strong evidence that molecular associations mediating DNAm-trait effects are predominantly due to vertical pleiotropy, even when only a limited number of IVs were available.

Supplementary Figure 15. MVMR sensitivity analysis excluding the top instrumental variable (pleiotropy sensitivity analysis). Mediation analyses were conducted for all DNAm-trait pairs with at least 3 exposure-associated IVs after excluding the top IV (i.e., exposure-associated IV with the lowest p-value; 1,590 DNAm-trait pairs). **a** \bar{MP} and 95% CI calculated on these pairs excluding the top IV in both the total and direct effect calculation. The slope is shown by the black line and the identity line by the blue line. **b** Corresponding MPs of traits grouped by physiological categories. The vertical dotted line corresponds to the mean MP across all DNAm-trait pairs and error bars represent the 95% CI. **c, d** Same analysis as in **a**

and **b**, respectively, but without excluding the top IV (same 1,590 pairs). **e** DNAm-to-transcript MR effects (α_{EM}) of the exposure-mediator pairs included in the mediation analyses of the 1,590 DNAm-trait pairs are shown before and after the exclusion of the top IV. **f** Conditional F-statistics calculated on the 1,590 DNAm-trait pairs before and after the exclusion of the top IV. Conditional F-statistics were on average 7.36 higher before excluding the top IV (two-sided t-test p-value = $5.37e-11$) pointing out that weak instrument bias was more present in the analyses where the top IV was missing.

Overall, the analyses show that excluding the top IV results in noisier MR estimates as a consequence of weaker instruments. While the top IV is crucial in getting robust molecular MR estimates, the analyses show that the remaining IVs support same effect size magnitudes and directionalities as the top IV.

Specifically, excluding the top IV significantly increased the MP estimated over all the DNAm-trait pairs (panel **a** vs **c**, $P_{diff} = 0.0228$). This difference is likely due to weak instrument bias as it was not present for pairs with $F > 10$ (845/1,590 pairs, MP = 40.9%, 95% CI: [29.3%, 52.4%] – top IV excluded; $P_{diff} = 0.48$). The estimated MP did not depend on the conditional F-statistic when all IVs were considered (Supplementary Fig. 10).

Supplementary Figure 16. Simulation analysis to assess the possibility of DNAm-to-transcript associations due to horizontal pleiotropy. For a significant DNAm-to-transcript MR association (P_{EM} , herein called P_{True}), we performed simulation tests ($N_{sim} = 100,000$) by randomly selecting eQTL-SNPs with identical multivariable eQTL effects in the region. Each simulated marginal eQTL effect estimate resulted in a random DNAm-to-transcript MR estimate ($P_{EM,j}$) from which we could derive the simulation p-value ($P_{sim} = \#(P_{EM,j} < P_{True})/N_{sim}$). **a** Comparison of the true (P_{True}) and random ($P_{EM,j}$) p-values from 100 DNAm-to-transcript MR estimates (the transcript outcome being the true and hypothetical transcript j from each simulation run, respectively). **b** QQ-plot of the simulation-based p-values P_{sim} .

The analysis shows that while MR p-values from hypothetical transcript effects are inflated, they are much less significant than the true p-values ensuring that horizontal pleiotropy is not at the root of observed methylation-expression causal effects.

3a – Specific answer

We now highlight in the manuscript, that the top variant is indeed important in obtaining robust and statistically well-powered MR estimates. However, the remaining variants still allow to recover mediation results provided that weak instrument bias is not an issue. Conditional F-statistics above 10 when the top IV was omitted were obtained for 845 out of 1,590 DNAm-trait pairs and for these pairs consistent MP estimates were obtained regardless of the inclusion of the top IV in the mediation analysis.

Thus, excluding the top IV does not change MP estimates (provided the conditional F-statistics check), however, the loss in statistical power is substantial given the generally low number of IVs available to infer molecular trait associations. (See detailed answers above.)

3b - Specific answer

As described in the method section (“Pleiotropy sensitivity analyses”, please see paragraph above), we followed this approach making small adaptations that take into account the LD structure of the region. Instead of distributing the marginal transcript effects to other SNPs in the region, we redistribute the multivariable SNP-transcript effects to better mimic the underlying biology and account for the LD in the region. Our approach also ensures that the marginal eQTL associations after pruning will be similar to the original one.

This simulation analysis demonstrates that while MR permutation p-values obtained from random significant eQTL SNPs are more significant than by chance (Supplementary Fig. 16b), they are still orders of magnitudes smaller than the true MR p-values (Supplementary Fig. 16a). It is worth keeping in mind that due to spatial constraints and the number of eQTL and mQTL SNPs, there is a real chance that they end up in being in LD, hence producing an inflated MR P-value.

4. I think I may be missing something about Figure 5. In Figure 5a, the mediated effect appears to be $-0.67 \times -0.13 = 0.087$. According to Figure 1 and the description, $MP = \theta_m / \theta_t = 0.087 / 0.11 = 0.79$. But the figure lists MP as 0.47. Is the MP given in the figure referring to something else? Is it an average over multiple methylation probes rather than only the probe shown in the figure? Same question about 5b in which it appears MP should be equal to 0.33.

There are two approaches of calculating mediated effects which explain the discrepancies between the numbers in Figure 5 and the numbers calculated by the reviewer.

In the first approach, direct effects are subtracted from total effects, which is also called the “difference in coefficients method” (Carter *et al.*, 2021). This is the approach we used as it is the most straightforward one: total effects are estimated in univariable and direct effects of the exposure on the outcome in multivariable MR analyses and subtracted from one another to derive the mediated effect. Here, effects from the exposure on the mediator are not involved *per se* (although calculated to choose mediators), neither are the effects from the mediators on the outcome (although obtained in the MVMR calculation).

The second approach is termed as the “product of coefficients method”. In this approach, the mediated effect is calculated explicitly by multiplying univariable MR estimates from the exposure on the mediator with the direct effects of the mediator on the outcome (from the MVMR analysis). Summing these effects across mediators yields the indirect effect. Since these two approaches use different input data (different instruments), they are not guaranteed to yield identical results, but our additionally performed analysis confirms high overall concordance between them (please see Supplementary Fig. 19 and explanations below).

In the example depicted in Figure 5a, the direct effect θ_D was 0.061 (Supplementary Data 1) which resulted in an MP of 0.47 ($MP = 1 - \theta_D / \theta_T = 1 - 0.0605 / 0.114 = 0.47$) which differs from the product of coefficient method which was employed by the reviewer.

To avoid confusion, we now introduce these two approaches of calculating mediation effects in our revised manuscript and changed the method section “Comparing mediation proportions” to “Estimating and comparing mediation proportions” explaining more explicitly how we calculate MPs throughout the manuscript. Furthermore, we demonstrate in an additional analysis that overall, both methods are concordant when applied to all 2,069 DNAm-trait pairs.

The new method section reads now as follows:

Mediation proportions (MPs) were estimated on sets of DNAm-trait pairs with significant total causal effects θ_T , either grouped by trait (if there were at least 10 such pairs within a given trait), trait category (e.g. hepatic traits, inflammatory traits/diseases) or combining all pairs together. MPs were then calculated by regressing θ_D on θ_T to estimate for the unmediated proportion, $\hat{\gamma}$, which after correcting for regression dilution bias (Eq. 4):

$$\hat{\gamma}_{cor} = \frac{\hat{\gamma}}{\sqrt{1 - \frac{\sum se^2(\hat{\theta}_T)}{\sum \hat{\theta}_T^2}}}$$

yielded $\widehat{MP} = 1 - \hat{\gamma}_{cor}$ for a defined set of DNAm-trait pairs, together with a standard error. For individual DNAm-trait pairs, we report the \widehat{MP} as $1 - \widehat{\theta}_D / \widehat{\theta}_T$, without providing its variance estimate since this would require individual-level data \cite{carter2021mendelian}.

In our approach, indirect effects θ_M are estimated by subtracting direct effects from total effects, which is also referred to as the difference in coefficients method \cite{carter2021mendelian}. Alternatively, the indirect effect can be estimated by the product of coefficients method \cite{carter2021mendelian}, where univariable MR estimates from the exposure on the mediator are multiplied with the direct effects of the mediator on the outcome (Eq. 3) and summed across mediators. Direct effects of the exposure on the outcome can then be obtained by the difference between the total and indirect effect. As demonstrated earlier \cite{carter2021mendelian}, the two approaches yield highly concordant results (Supplementary Fig. 19).

Supplementary Figure 19. Agreement between the product of coefficients and difference in coefficients methods to estimate direct and indirect effects. In the left panel, the agreement between

direct effects estimated from the multivariable Mendelian randomization regression (MVMR, difference in coefficients methods) and direct effects from the product approach is shown. In the product approach, exposure-to-mediator effects are multiplied with mediator-to-outcome direct effects and summed up across mediators to get the indirect effect. The direct effect is then calculated by subtracting this indirect effect from the total effect. The right panel shows the agreement between total effects obtained from the univariable MR regression and total effects reconstructed by summing the direct and indirect effects derived from the MVMR regressions. In the latter, the direct effect refers to the “Direct effect – MVMR” from the left panel and the indirect effect to the one obtained in the product approach. The R coefficient displayed in the plot area is the Pearson correlation coefficient (identity line is plotted in blue). Results are shown for all 2,069 DNAm-trait pairs colour-coded by the physiological category of the trait as defined in Supplementary Fig. 8.

5a. I was generally confused about when the estimated MP was an average and when it was a ratio. My understanding is that in Figure 3, the barplots are showing the average MP over all methylation probes with detectable mediation for a particular trait. Is there a reason to believe that all methylation probes should have the same mediated proportion? This MP isn't clearly described as an average, giving the impression that it is measuring a specific causal parameter. How much variability in the ratio MP estimates did you find within traits?

The estimated MP was always an average over at least 10 DNAm-trait pairs, apart from reporting examples of individual DNAm-trait pairs in the “Putative regulatory mechanisms of action” result section and when assessing factors underlying high MPs (Figure 4). We have not estimated individual MPs as the statistical power is too low to declare significantly non-zero individual MP estimate. In order to estimate the variance of the ratio statistic, we would need individual level data to compute the covariance between the numerator and the denominator (Carter et al., 2021). Assuming zero covariance leads to an overestimation of the variance of the ratio. Still, we have computed $Var(\widehat{MP})$ for all our DNAm-trait pairs, confirming that we are massively underpowered to declare individual MPs being significantly non-zero. Full details are provided in response to Q7.

The reviewer is correct that not all methylation probes are expected to have the same mediation proportion, this can vary for each DNAm-trait pair and they constitute an entire distribution. We are simply estimating the mean value of this distribution via regressing the mediated effects on the total effects. These mean values may differ by trait as we show in Fig 3b. Therefore, the variance of our MP estimator includes both the inaccuracy of each estimator and the variance of each (true) MP value across the different DNAm-trait pairs. Despite the fact that our MP estimator variance is inflated by heterogeneity in the individual MPs, it is still more powerful because it borrows strength from the large number of DNAm-trait pairs.

This has now been clarified in the method section “Estimating and comparing mediation proportions” in our answers to Q4 and Q7. Furthermore, we clarified in the result section “Application to 50 complex traits” that these come from a combined regression and not the average of ratios. The paragraph reads now as follows:

We first estimated the causal effects of DNAm probes on 50 complex traits, ranging from biomarkers indicative for diseases, such as low-density lipoprotein (LDL) and glucose levels, to diseases such as asthma and schizophrenia (Supplementary Table 1). DNAm-trait pairs with a significant total causal MR effect ($P_T < 1e-6$) were then further assessed to examine what fraction

of the DNAm→trait causal effect are mediated by transcripts in cis (Fig. 3a; Supplementary Fig. 1). Mediation analyses could be conducted for 2,069 pairs, for which at least 1 transcript was causally associated to the DNAm exposure (detectable mediation). First, we regressed $\hat{\theta}_D$ against $\hat{\theta}_T$ within each trait influenced by at least 10 DNAm probes while accounting for regression dilution bias (Eq. 6). \widehat{MP} s estimated for each of these 41 traits ranged from 18.0 to 78.0% (mean: 36.9%, 95% CI: [13.5%-60.3%]) with the trait with the highest \widehat{MP} being grip strength and the one with the lowest testosterone level (Fig. 3b). Regressing $\hat{\theta}_D$ against $\hat{\theta}_T$ for all pairs combined yielded an \widehat{MP} of 37.8% (95% CI: [36.0%-39.5%]) (Fig. 3c).

Figure 3. \widehat{MP} s for transcripts in cis mediating DNAm-to-trait effects. **a** Flowchart describing the selection of DNAm-trait pairs retained for mediation analyses. Among the total of 2,480,677 pairs

tested, 2,069 pairs (orange) with a significant total causal effect and at least 1 causally-associated transcript were assessed for mediation (**b** and **c**) – termed “detectable mediation”. **b** \widehat{MP} s by trait where error bars denote the 95% CI, and the grey vertical bar shows the mean \widehat{MP} across the traits. Only traits with ≥ 10 DNAm-trait pairs with detectable mediation are displayed (41 traits with number of evaluated pairs indicated in parentheses), colour-coded by their physiological category as defined in the legends of **c** and **d**. **c** Detectable mediation: All DNAm-trait pairs assessed in the mediation analyses (2,069) with traits being grouped into 10 physiological categories. The global \widehat{MP} in percentage with 95% CI is shown in the plotting area and individual category \widehat{MP} s in the legend. **d** Overall mediation: Same analysis as in **c**, but including all 2,623 DNAm-trait pairs with at least 1 transcript present in the cis region (incl. 554 pairs without any transcript causally linked to the DNAm site). For these additional pairs, the direct effect was set to the total effect.

5b. On the same theme, I think it would be worth clarifying that the 37.8% reported in the abstract is an average only over DNAm probes with any detectable mediation. This is very different from saying that 37.8% of methylation effects on the traits are mediated by transcripts.

Indeed, this is a good point. Throughout the manuscript, we no longer call mediation results that include DNAm-trait pairs with no causally associated mediator “adjusted mediation”, but “overall mediation”, and when we refer to DNAm-trait pairs with at least 1 mediator we call it “detectable mediation”. We changed the numbers in the abstract to report the overall mediation proportion (estimated MP for DNAm-trait pairs that could potentially be assessed for mediation). The abstract reads now as follows:

High-dimensional omics datasets provide valuable resources to determine the causal role of molecular traits in mediating the path from genotype to phenotype. Making use of quantitative trait loci (QTL) and genome-wide association studies (GWASs) summary statistics, we developed a multivariable Mendelian randomization (MVMR) framework to quantify the proportion of the impact of the DNA methylome (DNAm) on complex traits that is propagated through the assayed transcriptome. Evaluating 50 complex traits, we found that on average at least 28.3% (95% CI: [26.9%-29.8%]) of DNAm-to-trait effects were mediated through (typically multiple) transcripts in the cis-region. Several regulatory mechanisms were hypothesized, including methylation of the promoter probe cg10385390 (chr1:8'022'505) increasing the risk for inflammatory bowel disease by reducing PARK7 expression. The proposed integrative framework can be extended to other omics layers to identify causal molecular chains, providing a powerful tool to map and interpret GWAS signals.

Throughout the manuscript, we now clearly emphasize that the MP of 37.8% corresponds to DNAm-trait pairs with detectable mediation and early on introduce the distinction between MP results with detectable mediation and overall mediation. We changed the first result section “Overview of the methods” as follows:

MP estimates were then computed only for DNAm-trait pairs with significant Bonferroni-corrected $\hat{\theta}_T$ effects, grouped by trait, trait category and all pairs combined. We present MP results for DNAm-trait pairs with at least one mediator significantly associated to the exposure (“detectable mediation”), but also for pairs, including the ones without a significant causal effect on any potential transcript (“overall mediation”).

Furthermore, we added a scheme to show how DNAm-trait pairs in both scenarios were selected (new Figure 3a shown in the answer to Q5a). We also replaced the third panel in Figure 3 to display the MP when including DNAm-trait with no causally associated mediator (new Figure 3d panel; overall mediation analysis) and labelled the subplots 3c and 3d accordingly.

The first paragraph in the discussion was also changed accordingly:

We presented a framework to quantify mediation of complex trait-impacting effects through an omics layer and demonstrated its application to assayed blood-derived DNAm (exposure) and transcript levels (as mediator). Evidence for mediation of DNAm-to-trait effects through transcripts in cis was found to be at least 37.8% for the 2,069 evaluated DNAm-trait pairs with detectable mediation and 28.3% overall when considering the 2,623 pairs with significant total causal effects that could be assessed.

6. Should eq (5) be a t-distribution and not a standard normal? This equation contains a typo, I think one of the terms in the denominator should be $\hat{\gamma}_{cor}^2$.

Yes, strictly speaking, the statistic follows a t-distribution, but its degree of freedom is well over 50, thus a Gaussian approximation is very accurate. Thank you for spotting the typo in the equation, indeed the second term in the denominator should be $\hat{\gamma}_{cor}^2$ which has been corrected. The equation displays now as follows:

$$\frac{\hat{\gamma}_{cor}^{(1)} - \hat{\gamma}_{cor}^{(2)}}{\sqrt{se(\hat{\gamma}_{cor}^{(1)})^2 + se(\hat{\gamma}_{cor}^{(2)})^2}} \sim \mathcal{N}(0, 1)$$

7. In equation 4, if you are estimating the MP for a single methylation probe-trait pair, is it correct that $\hat{\gamma}$ is just the ratio θ_D/θ_T ? Do you never end up computing the MP for only a single probe (back to the confusion of question 5)?

If you did compute this value, $\hat{\gamma}$ would no longer be approximately normal and would not have moments as it is the ratio of two normals. It may be that you never encounter this problem because you are always averaging over many probes. That point is unclear to me.

We now explain more clearly that we only ever compute MPs as a ratio for single DNAm-trait pairs for the biological examples (please see the detailed answer to Q4 and Q5). In general, reported MPs are derived from the regression of θ_D on θ_T which provides the unmediated proportion $\hat{\gamma}$.

Standard errors of ratio MPs can be obtained by bootstrapping if individual-level data is available (Carter et al., 2021). This has been specified in the method section (see answer to Q4). However, since this data is not available to us, we do not report confidence estimates for individual MP ratios.

Statistically, it is possible to compute the variance of a ratio using the following formula (<https://www.stat.cmu.edu/~hseltman/files/ratio.pdf>):

$$\text{Var}(R/S) \approx \frac{1}{(\mu_S)^2} \text{Var}(R) + 2 \frac{-\mu_R}{(\mu_S)^3} \text{Cov}(R, S) + \frac{(\mu_R)^2}{(\mu_S)^4} \text{Var}(S) \quad (18)$$

$$= \frac{(\mu_R)^2}{(\mu_S)^2} \left[\frac{\text{Var}(R)}{(\mu_R)^2} - 2 \frac{\text{Cov}(R, S)}{\mu_R \mu_S} + \frac{\text{Var}(S)}{(\mu_S)^2} \right] \quad (19)$$

$$= \frac{(\mu_R)^2}{(\mu_S)^2} \left[\frac{\sigma_R^2}{(\mu_R)^2} - 2 \frac{\text{Cov}(R, S)}{\mu_R \mu_S} + \frac{\sigma_S^2}{(\mu_S)^2} \right] \quad (20)$$

We don't have information about the covariance term, but knowing that it is positive, its omission provides an upper bound. Using our notation to the use case of the θ_D/θ_T ratio gives:

$$\text{var} \left(\frac{\theta_D}{\theta_T} \right) = \frac{\widehat{\theta_D^2}}{\widehat{\theta_T^2}} \left[\frac{se(\widehat{\theta_D})^2}{\widehat{\theta_D^2}} + \frac{se(\widehat{\theta_T})^2}{\widehat{\theta_T^2}} \right]$$

We applied this formula to the 2,069 DNAm-trait pairs mediated by the top transcript compute the upper bound of the standard error of the θ_D/θ_T ratio. However, given the very large SEs obtained by this strategy (median of 0.3, see distribution in the figure below), we do not see the value in reporting it.

Reviewer #2

Thank you for the opportunity to review this interesting manuscript applying multivariable Mendelian randomisation to quantify the role of transcript levels in mediating DNA methylation on outcomes. The manuscript has a number of strengths including comprehensive analyses and clarity. It also offers an example of approaches that can be used to better understand the pathway from genotype to disease via multiple layers of intermediate molecular traits. Below, I list a number of questions that I have regarding the manuscript.

How the instruments are selected could be made clearer. When I first read the manuscript, I thought that instruments for the exposure (DNAm) were selected rather than instruments for both the exposure and the mediator (transcript levels). Reading the method section, I believe that the latter is true, which makes more sense, maybe this can be made explicit earlier on, for example near figure 1.

We would like to thank the reviewer for the positive assessment of our work and the constructive comments which we addressed in the following.

We redrew the scheme in Figure 1 to make it clearer that instruments were selected to be associated with either the DNAm exposure or the transcript mediators and adapted the legend accordingly.

Figure 1. Overview of the MVMR design to quantify mediation of complex traits through DNAm and transcripts. Genetic instruments (SNPs) are selected to be directly and significantly associated (dashed arrows) with either the exposure (DNAm) or any mediator k (transcript in *cis*). The total effect θ_T (blue-green dotted arrow) of the exposure on the outcome (complex trait) is estimated in a univariable MR analysis based on exposure-associated SNPs only. The direct effect θ_D is estimated in a MVMR analysis on all valid instruments. The mediation effect θ_M results from the difference between θ_T and θ_D , and allows to calculate the mediation proportion (MP). The genetic effect sizes β on the exposure, mediator and

outcome come from m/eQTL and GWAS summary statistics, respectively. Transcripts were required to be causally associated to the DNAm-exposure to be included as mediators.

In classical mediation analyses, many researchers tend to preselect significant relationships to be modelled in the mediation framework. The selection is either transparent or, what is worse, does not appear in the final article. This is obviously an issue: if you compute a 100 correlations, 5 of them will be significant by chance, and any mediation model applied to these variables is likely to yield significant mediation effects were none are present. I know the authors are aware of this issue, but I think their approach could be reported more clearly. For example, they pre-filter DNAm probes that are causally related to the outcome, and transcripts that are causally related to DNAm.

“MP estimates were then computed only for DNAm-trait pairs with significant Bonferroni-corrected θ effects, grouped by trait, trait category and all pairs combined. We further filtered out DNAm-trait pairs whose exposures have no significant causal effect on any potential transcript as the link”

The selection is drastic, e.g. around 2000 DNAm causally related to outcomes out of 1 million DNAm to outcomes relationships if I got this properly. This should naturally lead to more significant mediation findings in this subset and likely exaggerate the MP (mediated proportion)? The authors propose a number of sensitivity analyses, which may already tackle this issue.

This is an excellent point. We have explored the impact of preselection threshold (P_{EM}) on the MP: milder thresholds would indeed allow overestimation of the MP. However, potential overestimation is very mild in our biologically realistic settings (see Supplementary Figure 13-14). In addition, we have also varied the number of mediators up to 10, but it did not impact the results either (see Supplementary Figure 7). The overfitting phenomenon invoked by the reviewer is much less of an issue in our settings, since we are investigating mediation in *cis* only. This largely limits the number of mediators (genes), which are often correlated to each other, reducing the effective number even further. In addition, we can consider only mediators with sufficient number of instruments, which leads to further drop in this number.

However I would appreciate:

1. A clear reporting in one place in the manuscript of all numbers involved at the different stages of filtering (i.e. initial number of probes; number after filtering for causal relationship with the outcome; initial number of transcripts etc...). I think most of these numbers are already present in the manuscript but in different places. Maybe a comprehensive flowchart would be useful.

We thank the reviewer for this suggestion. We now added a comprehensive flowchart to Figure 3 (new panel **a**) reporting the initial number of DNAm probes and DNAm-trait pairs tested, initial number of transcripts, remaining number of DNAm-trait pairs with significant Bonferroni-corrected θ_T effects, number of pairs with mediators in the region and those for which mediators were causally associated, and the number of pairs for which no mediation analysis could be conducted because of insufficient number of instrumental variables. This flowchart replaces our previous Supplementary Figure 16 where some of these numbers have been previously reported.

Figure 3. \widehat{MP} s for transcripts in *cis* mediating DNAm-to-trait effects. **a** Flowchart describing the selection of DNAm-trait pairs retained for mediation analyses. Among the total of 2,480,677 pairs tested, 2,069 pairs (orange) with a significant total causal effect and at least 1 causally-associated transcript were assessed for mediation (**b** and **c**) – termed “detectable mediation”. **b** \widehat{MP} s by trait where error bars denote the 95% CI, and the grey vertical bar shows the mean \widehat{MP} across the traits. Only traits with ≥ 10 DNAm-trait pairs with detectable mediation are displayed (41 traits with number of evaluated pairs indicated in parentheses), colour-coded by their physiological category as defined in the legends of **c** and **d**. **c** Detectable mediation: All DNAm-trait pairs assessed in the mediation analyses (2,069) with traits being grouped into 10 physiological categories. The global \widehat{MP} in percentage with 95% CI is shown in the plotting area and individual category \widehat{MP} s in the legend. **d** Overall mediation: Same analysis as in **c**, but including all 2,623 DNAm-trait pairs

with at least 1 transcript present in the cis region (incl. 554 pairs without any transcript causally linked to the DNAm site). For these additional pairs, the direct effect was set to the total effect.

2. A more thorough discussion of how filtering at different stages might impact the results, not only for MPs but also potentially false positives. The authors have done multiple testing correction at different stages, but does that fully account for the problems mentioned above (i.e. only carrying forward significant bivariate relationships in the mediation model).

Throughout the manuscript, we now emphasize more clearly that our previous “main mediation” results (37.8% through transcripts in *cis*) relate to DNAm-trait pairs with significant total effects and with detectable mediation (i.e., at least one causally associated mediator; 2,069 pairs). In our revised manuscript, we report as main mediation results the ones derived by considering all DNAm-trait pairs with significant total effects that could potentially be assessed for mediation (even if no transcript mediator has been detected; 2,623 pairs) and refer to them as “overall mediation”. This contrasts with the “detectable mediation”, which refers to pairs with at least one mediator.

We changed the abstract accordingly:

*High-dimensional omics datasets provide valuable resources to determine the causal role of molecular traits in mediating the path from genotype to phenotype. Making use of quantitative trait loci (QTL) and genome-wide association studies (GWASs) summary statistics, we developed a multivariable Mendelian randomization (MVMR) framework to quantify the proportion of the impact of the DNA methylome (DNAm) on complex traits **that is propagated through the assayed transcriptome**. Evaluating 50 complex traits, we found that on average at least **28.3% (95% CI: [26.9%-29.8%])** of DNAm-to-trait effects were mediated through (typically multiple) transcripts in the *cis*-region. Several regulatory mechanisms were hypothesized, including **methylation of the promoter probe cg10385390 (chr1:8'022'505) increasing the risk for inflammatory bowel disease by reducing PARK7 expression**. The proposed integrative framework can be extended to other omics layers to identify causal molecular chains, providing a powerful tool to map and interpret GWAS signals.*

Furthermore, we replaced the third panel in Figure 3 to display the MP when including DNAm-trait pairs with no causally associated mediator (new Figure 3d panel; overall mediation analysis → shown in the answer to Q1).

As pointed out in Carter et al., 2021 (Section: “Small total effects”), mediation results in the absence of significant or weak total effects “should be interpreted with caution”. The authors of this paper showed in simulation studies that mediation proportions estimated in such scenarios have very large standard deviations and may thus be very inaccurate. In addition, when there is no total effect because of direct and indirect effects cancelling each other out, estimates “are prone to inflated type 1 errors (i.e., false positive results).” Given that mediation proportions are ill-defined when the exposure-outcome relationship is not significant, we focus on DNAm-trait pairs for which there is a significant total effect.

3. A related comment is on the accuracy, or even the relevance of the MPs that are reported. As the authors mention, MPs are very dependent on technical issues. For example, MPs are higher when the transcript is closer to the DNAm site. But transcripts are imperfectly measured (i.e. as

the authors mention available data is a magnitude lower than the transcriptome). Can we imagine that if we had the perfect dataset, MPs would be close to 1 all the time? In brief, there is a possibility that the authors choices (selection, see above) might overestimate MPs, but that the limitations of the datasets, might underestimate MPs. If this is the case, how reliable and meaningful are the MPs results? I wonder whether the authors need to insist so much on MPs, which are currently an important focus of the manuscript. Their approach seems more interesting to me to discover pathways (of which the authors give a few examples) rather than to estimate abstract quantities like MPs?

We report mediation proportions as a main quantifiable entity for several reasons. First, it provides an estimate to what extent we can leverage current large-scale mQTL and eQTL datasets to find molecular pathways of the scheme DNAm-transcript-trait. We agree with the reviewer that discovering pathways is of greater biological relevance than MPs. However, due to data (un)availability, we can only explore *cis* mediation, which largely reduces the interrogation of complex pathways. Furthermore, given the limitations of the sample size, we are underpowered to reliably identify statistically significant individual mediation events. The obtained average MP of 28.3% is only a lower bound, but still very useful to know, as it is the first estimate of its kind. Other studies, e.g. estimating the proportion of heritability mediated by assayed gene expression (Yao et al., 2020, *Nature Genetics*. <https://doi.org/10.1038/s41588-020-0625-2>), also established such lower bounds with similar caveats and were well received. Also, having a quantified MP allows to compare different settings and to identify factors that predict when mediation through transcripts predominantly explains the pathway from methylation to phenotype. Furthermore, we believe that the link between higher mediation by closer transcript is not a technical artefact, but makes sense biologically (as it is the case for SNP - gene expression relationship). Similarly, in our new version, we found that the mediation is higher when DNAm is decreasing transcript levels (please see our detailed answer to Q4). Moreover, even if MP is underestimated due to measurement/technical issues, comparing MPs across traits, or in different biological context remains meaningful. Finally, we believe that the true mediation proportion cannot be 1, since it would mean that methylation has no other role than modulating gene expression, which is highly unrealistic.

In addition, we extended our simulation analyses to show that MP estimates are dependent on the exposure and mediator heritabilities. More specifically, even if transcripts are measured perfectly, but have a low heritability (i.e., genetic component that explains the variability in transcript levels) then we are likely to underestimate MPs. These results have been added to the new Figure 2 and we added a paragraph to the result section “Simulation results” which reads as follows:

Furthermore, we investigated weak instrument bias of both exposure- and mediator-associated IVs. When mediator-associated IVs were weak (i.e., low direct mediator heritabilities ($h_{M,direct}^2$; Methods), a high variability and significant underestimation of the \widehat{MP} was observed (Fig. 2c). In case of low mediator heritability, the conditional F-statistic of the exposure was also below the critical threshold of < 10 (Methods) indicating weak instruments. Similarly, for low exposure heritability (h_E^2), underestimated \widehat{MP} s were obtained, even in case of high conditional F-statistic (> 120 ; Fig. 2d). Additional simulation studies with more polygenic exposures and increased number of relevant mediators N_{med} for different exposure and mediator heritabilities corroborated the findings of underestimated \widehat{MP} s in case of weak instruments (Supplementary Fig. 7).

Figure 2. Simulation results to assess the bias in estimated MPs (\widehat{MP}) in real data settings. a Influence of the mediator sample size on the estimated \widehat{MP} (orange) and number of selected mediators ($N_{med,sig}$, blue). **b** Influence of the mediator selection threshold P_{EM} . **c, d** Sensitivity of \widehat{MP} s in settings of weak instruments, simulated by low mediator ($h^2_{M,direct}$) and exposure (h^2_E) *cis*-heritabilities, respectively. Conditional F-statistics of the exposure allow to test for weak instruments bias (critical values are defined at a threshold of F-statistic < 10). For a given parameter setting, 500 exposure-outcome pairs were simulated on which an \widehat{MP} and 95% CI (error bars) were estimated. The true MP of the model was 0.35 (horizontal orange lines), and the true number of relevant mediators N_{med} was 2 (horizontal blue lines) among a set of 12 potential mediators (20 in **b**).

4. Also, the authors selected only probes with significant effects on outcomes. However, a mediated effect can be observed even in the absence of a univariate effect because of suppression effects (e.g., two indirect paths via transcript levels in opposite direction and summing up to 0). The authors mentioned that the relationships between DNA methylation and transcript levels was not always negative as expected, in how many cases did they have different signs in the DNAm \rightarrow transcript paths for a give probe (e.g. 3 transcript levels, 1 related positively to DNAm and 2 negatively) and does this affect their findings.

This is a very good point; we thank the reviewer for mentioning it. First, we would like to specify that, in our definition, mediation only refers to the fact that a non-zero causal effect is partially acting through other entities. If the total effect is zero, there is nothing to be mediated. Also, if the total causal effect is zero because the direct and indirect causal effects cancel each other out, mediation proportion would be undefined (division by zero). If direct and the (total) indirect effect(s) are of opposite sign, the mediation proportion could be arbitrarily large in magnitude and sign, which is less meaningful. Still, as long as the total effect is significantly non-zero in these situations, these scenarios are detectable in our analysis (and occasionally picked up), but overall negligible (otherwise the average MP would not be positive).

The other excellent point of the reviewer is the possible situation, whereby multiple indirect effects disagreeing in sign exist. We now performed an additional analysis where we stratified DNAm-trait pairs by the effect sign the DNAm has on the transcript level. The results are summarized in a new Table 1, where we report the number of times the effect sign has been negative, positive or a mixture of both (additionally stratified by the number of mediators). For each stratum, the corresponding MP has been calculated and reported as well. The numbers clearly show that when there is only a single transcript mediator, then mediation is more pronounced if DNAm is decreasing expression rather than increasing it. When the DNAm site was affecting multiple transcripts in mixed directions, we could not observe that these effects were cancelling each other out, although a higher number of mediators in these situations (more mediators → higher MPs) was confounding these results.

These analyses have been added to a new paragraph in the “Determining factors of mediation proportions” result section and reads as follows:

Furthermore, we investigated whether \widehat{MP} s are dependent on the DNAm→transcript causal effect directions following the logic of a recent DNAm-transcript correlation study \cite{ruiz2022identification}. To this end, we stratified DNAm-trait pairs by the α_{EM} sign and number of mediators (Table 1). If there was only a single mediator, \widehat{MP} s were significantly higher if the DNAm was decreasing expression ($P_{diff} = 3.49e-8$). This is consistent with the observation that negative effects α_{EM} were larger than positive ones and the positive correlation between α_{EM} magnitudes and high \widehat{MP} s (Fig. 4d). When there were multiple mediators, most DNAm sites had negative effects on some transcripts and positive effects on others. These bivalent DNAm probes exhibited the highest \widehat{MP} s ($\widehat{MP} = 53.9\%$ (95% CI: [51.2%-56.5%])) - a consequence of being causally associated to more mediators than average (5.01 vs 3.31), with N_{med} being a strong predictor for high \widehat{MP} s (Fig. 4e). Combining DNAm-trait pairs with single and multiple mediators, but with consistent negative or positive α_{EM} values, the observation of higher \widehat{MP} s when DNAm was decreasing transcript levels persisted ($P_{diff} = 0.020$).

Table 1: Exposure-to-mediator effect direction and number of mediators explaining MPs. DNAm-trait pairs were stratified by the number of mediators (N_{med} ; “mono” if $N_{med} = 1$ and “multi” if $N_{med} > 1$) and by the exposure-to-mediator α_{EM} causal effect sign (“Negative” and “Positive” for DNAm decreasing and increasing transcript levels, respectively, and “Bivalent” if a given DNAm site is affecting transcript levels in both directions). For each stratum, the number of DNAm-trait pairs (N), the estimated mediation proportion (\widehat{MP}) and mean N_{med} is shown.

	Negative	Positive	Bivalent	Total
Mono	N = 370 (17.9%) $\widehat{MP} = 20.8\%$ (95% CI: [17.3%-24.2%])	N = 276 (13.3%) $\widehat{MP} = 7.97\%$ (95% CI: [5.04%-10.9%])	0, by definition	N = 646 (31.2%) $\widehat{MP} = 14.7\%$ (95% CI: [12.3%-17.1%])
Multi	N = 239 (11.6%) $\widehat{MP} = 42.8\%$ (95% CI: [38.3%-47.2%], mean(N_{med}) = 3.01)	N = 207 (10.0%) $\widehat{MP} = 42.8\%$ (95% CI: [38.3%-47.3%], mean(N_{med}) = 2.84)	N = 977 (47.2%) $\widehat{MP} = 53.9\%$ (95% CI: [51.2%-56.5%], mean(N_{med}) = 5.01)	N = 1423 (68.8%) $\widehat{MP} = 50.3\%$ (95% CI: [48.2%-52.4%], mean(N_{med}) = 4.35)
Total	N = 609 (29.4%) $\widehat{MP} = 27.7\%$ (95% CI: [24.8%-30.5%], mean(N_{med}) = 1.79)	N = 483 (23.3%) $\widehat{MP} = 22.8\%$ (95% CI: [19.8%-25.8%], mean(N_{med}) = 1.79)	N = 977 (47.2%) $\widehat{MP} = 53.9\%$ (95% CI: [51.2%-56.5%], mean(N_{med}) = 5.01)	N = 2,069 (100%) $\widehat{MP} = 37.8\%$ (95% CI: [36.0%-39.5%], mean(N_{med}) = 3.31)

5. The authors look at reverse causation by implementing Steiger filtering for both the DNAm-> outcome and DNAm -> transcript, which is important. But their theoretical model is still DNAm -> transcripts -> outcome rather than transcript -> DNAm -> outcome. Maybe it could be made more explicit for the reader less familiar with basic biology, why this is the chosen model.

We explain now more explicitly that we chose to investigate the commonly assumed model of DNAm -> transcript -> outcome rather than the transcript -> DNAm -> outcome model. This has been added to the “Univariable and multivariable Mendelian randomization” method section when covering the Steiger filtering part:

As our MVMR model assumes a chain of causal effects from the exposure to the mediator and then to the outcome, we conducted several Steiger filtering steps to reduce biases due to reverse causation. Although it has been proposed that DNAm could be a consequence of gene expression in the same locus cite{gutierrez2013passive}, our model investigates the commonly assumed concept of DNAm regulating gene expression. In addition to meeting the Steiger criterion described above, exposure-associated IVs were required to pass that same threshold t_{rev} of no larger mediator than exposure effects for each of the mediators M_k .

6. If we have three paths in the multivariate model: a (DNAm -> transcript) b (transcript -> outcome) and c' (direct effect), with c being the univariate effect (DNAm -> outcome). In classical mediation analysis, there are two ways of computing the proportion mediated: (i) as the authors did $(1-c')/c$ or (ii) $a*b/c$. Can the authors say: (i) is it possible to use the second approach, which is usually preferred in classical mediation analysis, within MVMR; (ii) why did they choose the first approach over the second. Related to that is the question of whether estimates actually sum up to 1? I.e. given that you have different instruments for the exposure and the mediator, if you were to compute a and b, would $a*b + c' = c$.

Indeed, there are two ways of computing direct and indirect effects in mediation analyses which have previously been called the “difference in coefficients method” and “product of coefficients method” (Carter *et al.*, 2021). We specified this now in the method section and conducted additional analyses to compare the two approaches. (i) Yes, the second approach, product of coefficients method (indirect effect = $a*b$), can be used in the mediation analysis. However, since not the exact same instruments are used as in the “difference in coefficients method”, identical results are not guaranteed. However, our analysis confirms high overall concordance between them (please see Supplementary Fig. 19 and details below). (ii) When using the

product of coefficient method, the mediated effect is calculated explicitly by multiplying univariable MR estimates from the exposure on the mediator with the direct effects of the mediator on the outcome (from the MVMR analysis) and summing up these effects across mediators. In comparison, the difference in coefficients method, which we use, is more straightforward: direct effects of the exposure on the outcome are obtained from the MVMR calculation and subtracted from the total effect. Here, effects from the exposure on the mediator are not involved *per se* (although calculated to choose mediators), neither are the effects from the mediators on the outcome (although obtained in the MVMR calculation). Given the simplicity, this approach was chosen over the former one.

We conducted an additional analysis to see whether the estimates from both approaches sum up to 1 (i.e., $a*b + c' = c$). Although slight deviations can be observed for some DNAm-trait pairs, there is overall a very high concordance (Supplementary Fig. 19b).

The explicit presentation of the two methods and the results comparing them have been added to the new “Estimating and comparing mediation proportions” method section (former “Comparing mediation proportions” section). The added paragraph reads now as follows:

In our approach, indirect effects θ_M are estimated by subtracting direct effects from total effects, which is also referred to as the difference in coefficients method \cite{carter2021mendelian}. Alternatively, the indirect effect can be estimated by the product of coefficients method \cite{carter2021mendelian}. Here, univariable MR estimates from the exposure on the mediator are multiplied with the direct effects of the mediator on the outcome (Eq. 3) and summed across mediators. Direct effects of the exposure on the outcome can then be obtained by the difference between the total and indirect effect. As demonstrated earlier \cite{carter2021mendelian}, the two approaches yield highly concordant results (Supplementary Fig. 19).

Supplementary Figure 19. Agreement between the product of coefficients and difference in coefficients methods to estimate direct and indirect effects. In the left panel, the agreement between direct effects estimated from the multivariable Mendelian randomization regression (MVMR, difference in coefficients methods) and direct effects from the product approach is shown. In the product approach, exposure-to-mediator effects are multiplied with mediator-to-outcome direct effects and summed up across

mediators to get the indirect effect. The direct effect is then calculated by subtracting this indirect effect from the total effect. The right panel shows the agreement between total effects obtained from the univariable MR regression and total effects reconstructed by summing the direct and indirect effects derived from the MVMR regressions. In the latter, the direct effect refers to the “Direct effect – MVMR” from the left panel and the indirect effect to the one obtained in the product of approach. The R coefficient displayed in the plot area is the Pearson correlation coefficient (identity line is plotted in blue). Results are shown for all 2,069 DNAm-trait pairs colour-coded by the physiological category of the trait as defined in Supplementary Fig. 8.

7. Binary outcomes like schizophrenia are mentioned but I have seen no mention of whether there are specific considerations to take into account when using binary outcomes (i.e. the outcome gwas would include log odds).

We now made a note in the method section that in case of binary outcome GWASs, log-odds ratios were used as effect sizes and that results should be interpreted on the liability scale. The corresponding paragraph in the “Omics and trait summary statistics” section reads now as follows:

Remaining GWAS data came mostly from case/control studies made available by the consortium of the respective disease. For binary outcome traits, log-odds ratios were used as effect sizes and results should be interpreted on the liability scale.

8. Can the authors mention more specifically, for example in the methods section of the discussion, if there are any novel statistical aspects in their methods, or if it is applying existing MVMR methods.

We now added explicitly in the concluding part of the discussion section, that our mediation framework makes use of existing MVMR methods. It reads now as follows:

To conclude, by adapting existing MVMR mediation techniques to molecular exposures and mediators, we quantified the causal connectivity between DNAm and transcript levels, and their importance in shaping complex traits. Overall, we found solid evidence that at least a third of DNAm-to-complex trait effects are mediated by transcripts in cis. Our integrative omics framework can be extended to other omics-GWAS combinations and provide a powerful tool for mapping GWAS signals to biological pathways and prioritizing functional follow-up experiments.

9. The authors computed heterogeneity statistics to test for pleiotropy. Is there more sophisticated way to do this. I assume that MR Egger would not work because of the number of instruments? But any other methods, for example to remove outliers, like MR-PRESSO?

MR Egger or MR-PRESSO have been routinely used for the univariable MR analyses, but given the limited number of IVs in molecular traits MR analyses they are less appropriate for our MVMR framework. Instrument exclusion based on heterogeneity has its own biases in situations when the number of IVs is low.

However, to circumvent these limitations, we added additional sensitivity analyses (tailored to our use case) to estimate the expected amount of horizontal pleiotropy. First, we repeated the MVMR analyses excluding the top IV variant. This allows to determine whether the remaining

IVs recover the same MR estimates which allows to demonstrate that MR and MVMR estimates are not driven by strong outliers. We included these results in the section “MVMR sensitivity analyses” by adding the following paragraph:

Finally, we conducted sensitivity analyses to determine whether significant MR associations were due to horizontal pleiotropy. Regulatory pathways between DNAm exposure probes and transcript mediators were assessed in cis. As such, SNPs in LD with significant QTLs for both quantities could give rise to an association merely because of horizontal pleiotropy (i.e., due to random overlap between cis-QTLs in close vicinity), an issue further exacerbated by the fact that molecular omics entities generally have fewer associated IVs than complex traits. To assess whether mediation results are only based on a single strong genetic instrument, we repeated the mediation analysis excluding the top IV (i.e. exposure-associated IV with the lowest p-value) from both the total effect θ_T and direct effect θ_D calculations (Methods). The results show that while MR effect estimates remain concordant in magnitude and effect direction, the estimates are noisier due to the much weaker instruments (significantly lower F-statistics; two-sided t-test: $P = 5.37e-11$; Supplementary Fig. 15). MP estimates were also somewhat higher when the top IV was excluded ($\widehat{MP} = 47.3\%$ (95% CI: [38.4%-56.2%]); $P_{diff} = 0.023$; Supplementary Fig. 15), however, this was no longer the case when controlling for conditional F-statistics > 10 ($\widehat{MP} = 40.9\%$ (95% CI: [29.3%-52.4%]); $P_{diff} = 0.48$). Additionally, we performed simulation analyses to assess the possibility of significant DNAm-transcript associations caused by cis-mQTL and -eQTL signals being in LD (Methods). The analysis shows that randomly picked eQTL-SNPs in the region result in slightly inflated, but much weaker MR associations than using the original eQTL data (Supplementary Fig. 16). The results indicate that by-chance LD between cis-mQTLs and -eQTLs can yield false positive findings, but those signals are substantially weaker than the ones observed in real data. In other words, mQTL and eQTL IVs are in much higher LD than expected by chance.

Overall, these sensitivity analyses showed that the estimated MPs remain robust when removing DNAm-trait pairs that potentially violate MVMR assumptions, while also suggesting that the set P_{EM} threshold of 0.01 may lead to underestimated MP estimates. Finally, we found strong evidence that molecular associations mediating DNAm-trait effects are predominantly due to vertical pleiotropy, even when only a limited number of IVs were available.

Supplementary Figure 15. MVMR sensitivity analysis excluding the top instrumental variable (pleiotropy sensitivity analysis). Mediation analyses were conducted for all DNAm-trait pairs with at least 3 exposure-associated IVs after excluding the top IV (i.e., exposure-associated IV with the lowest p-value; 1,590 DNAm-trait pairs). **a** \bar{MP} and 95% CI calculated on these pairs excluding the top IV in both the total and direct effect calculation. The slope is shown by the black line and the identity line by the blue line. **b** Corresponding MPs of traits grouped by physiological categories. The vertical dotted line corresponds to the mean MP across all DNAm-trait pairs and error bars represent the 95% CI. **c, d** Same analysis as in **a**

and **b**, respectively, but without excluding the top IV (same 1,590 pairs). **e** DNAm-to-transcript MR effects (α_{EM}) of the exposure-mediator pairs included in the mediation analyses of the 1,590 DNAm-trait pairs are shown before and after the exclusion of the top IV. **f** Conditional F-statistics calculated on the 1,590 DNAm-trait pairs before and after the exclusion of the top IV. Conditional F-statistics were on average 7.36 higher before excluding the top IV (two-sided t-test p-value = $5.37e-11$) pointing out that weak instrument bias was more present in the analyses where the top IV was missing.

Overall, the analyses show that excluding the top IV results in noisier MR estimates as a consequence of weaker instruments. While the top IV is crucial in getting robust molecular MR estimates, the analyses show that the remaining IVs support same effect size magnitudes and directionalities as the top IV.

Specifically, excluding the top IV significantly increased the MP estimated over all the DNAm-trait pairs (panel **a** vs **c**, $P_{diff} = 0.0228$). This difference is likely due to weak instrument bias as it was not present for pairs with $F > 10$ (845/1,590 pairs, MP = 40.9%, 95% CI: [29.3%, 52.4%] – top IV excluded; $P_{diff} = 0.48$). The estimated MP did not depend on the conditional F-statistic when all IVs were considered (Supplementary Fig. 10).

10. The authors say in the discussion

“Their results indicated that colocalization between DNAm sites and complex traits were typically due to horizontal pleiotropy as opposed to DNAm mediating the GWAS effect. These results confirm that colocalization analysis is not necessarily able to detect causal relationships”

I’m not sure I understand here “DNAm mediating the GWAS effect”. This doesn’t appear to be the same question as it’s about whether DNAm mediates SNP effects on the outcome rather than whether transcripts mediate DNAm effects on outcome. How is that reference relevant to the current work. Also, isn’t that the case that colocalization is better at identifying relationships that are due to LD rather than true biological pleiotropy (i.e. same variant at play), which MR doesn’t really do? I think it would be important to include a lengthier and more nuanced discussion of the respective limitations and strength of the different methods that the authors mention. This will allow the authors to highlight the interest of their approach and how it complements existing ones.

Indeed, the formulations were not well chosen, and we agree with the reviewer’s observations. Hence, we have substantially changed this paragraph in the discussion which reads now as follows:

Statistical methods to integrate GWAS with omics data have seen a surge in recent years. Namely, colocalization methods based on a single genetic signal or corroborated by a secondary one, as well as methods supported by the SMR HEIDI statistic have been previously used in the study of DNAm-to-complex trait effects \cite{giambartolomei2014bayesian, min2021genomic, wu2018integrative}. In the most recent publication of the GoDMC consortium, the former strategy was applied to systematically evaluate DNAm and GWAS co-localizing signals and compared them to MR \cite{min2021genomic}. This revealed a relatively poor overlap between colocalization and MR results, for several reasons, as both approaches have their weaknesses in detecting causal relationships. The major weakness of colocalization analysis is that it cannot detect directionality and does not estimate causal effect size. Colocalization of local association signals of two traits may be due to causal effects in either direction, common local confounder effect (e.g. shared regulatory mechanism) or causal markers in very high LD. Lack of colocalization can happen even if there is a true causal relationship, but there are additional associations impacting only the outcome trait. On the other hand, the major weakness of MR is that it may falsely detect a causal relationship when the causal variants for each trait in the locus

are in reasonably high LD. The comparison of these two approaches is out of the scope of this work, but to explore the above-mentioned weakness in our study, we performed simulations tailored to detect by-chance overlaps in the association signals for methylation and gene expression (see pleiotropy sensitivity analyses for details). These analyses indicated that indeed elevated false positive rates are expected for MR, but the resulting MR p-values under the null are much less significant than the ones observed for real methylation-transcript data.

11. “The original method additionally includes the phenotypic correlation matrix between the mediators, however, due to the lack of these data, we used the identity matrix instead.”

The identity matrix would assume a correlation of zero between mediators? Isn't that unlikely? The simple fact that different transcripts are influenced by the same DNAm site would generate a non-null phenotypic relationship between them. There must be accessible data providing a general sense of how correlated expression data are, which would provide a better approximation?. Or maybe a sensitivity analysis using a different correlation matrix.

We thank the reviewer for the suggestion. There are two types of external data that would be necessary to properly conduct the conditional F-statistic computation: gene-gene expression correlation matrix and the methylation-gene expression correlations. Unfortunately, we have no access to the latter, but we conducted a sensitivity analysis where we included the gene-gene correlation matrix. We restricted the analysis to pairs with at least 2 mediators and for which we had correlation data available for at least half of them. Correlations between transcripts were based on RNAseq data from the CoLaus study. The results show that incorporating the correlation matrix between mediators is slightly increasing the conditional F-statistics. Therefore, assuming zero-correlations between mediators provides more conservative estimates.

The method subsection “Conditional F-statistic” reads now as follows:

Conditional F-statistics of the exposure were calculated following the approach of Sanderson et al. \cite{sanderson2021testing}. This method involves the regression of the exposure on the mediators based on the IV effect sizes on each of these quantities. The residuals of this regression are then used to derive the conditional F-statistic. The original method additionally includes the phenotypic correlation matrix between the exposure and mediators, which we omitted by default due to the lack of these data and thus used the identity matrix instead. However, as a sensitivity analysis, we calculated conditional F-statistics incorporating the phenotypic correlations between transcript mediators. Transcript correlations were calculated on RNAseq data from the Cohorte Lausannoise (CoLaus) based on 555 samples \cite{sonmez2021untargeted}. Transcript correlations could be estimated for 19,517 transcript of which 15,021 overlapped with the eQTL dataset (Methods: Omics and trait summary statistics) \cite{vosa2021large}. We then calculated conditional F-statistics that included mediator correlations for all DNAm-trait pairs with at least 2 mediators and for which at least half of them had available correlation data. Conditional F-statistics > 10 allow to reject the null hypothesis that the IVs are too weak to reliably estimate the multivariable effect of the exposure in the presence of the mediators.

The corresponding results are referenced in the “MVMR sensitivity analyses” section as follows:

Pairs with F-statistics < 10 (N=1,008) had significantly more mediators (4.32 vs 2.35, two-sided t-test: $P = 2.13e-64$), but not a significantly higher \widehat{MP} (mean: 40.9%, 95% CI: [37.8%-44.0%]; $P_{diff} = 0.08$; Supplementary Figs. 10-11).

Supplementary Figure 11. Conditional F-statistics with and without the correlation matrix between mediators. Conditional F-statistics with transcript-transcript correlations were calculated for all DNAm-trait pairs with at least 2 mediators and for which at least half of them had available correlation data. In total, 1,208 pairs were assessed with the mean F-statistics being 13.55 with the correlation matrix and 12.85 without. The slope is plotted in black and the identity line in blue.

Minor:

12. In Figure 1, arrows could be clearer. For example, the θM should have an arrow leading to transcription and another arrow from transcription to outcome, consistent with the classical way of representing mediation.

We thank the reviewer for the suggestion. We now adapted Figure 1, to include an arrow from the DNAm-exposure to the transcript-mediators and from the transcript-mediators to the outcome. Please see the new Figure 1 in the answer to Q0.

13. The authors say:

“Surprisingly, 46.6% of significant DNAm-to-transcript effects were 233 of positive sign (i.e. DNAm increasing transcription), particularly so when the DNAm site was situated in the gene body (enrichment p-value = $2.15e-10$). This observation is in line with previous genome-wide methylation and gene expression association studies which reported high fractions of positive correlations (30-35%)^{36,3}”

How is it surprising if in line with prior investigations? Maybe change the phrasing.

We rephrased the sentence and referenced a new study from 2022 where the fraction of positive correlations was 41.2%. The paragraph reads now as follows:

Additionally, this mediation framework allowed to quantify the causal connectivity and directionality between DNAm and transcript levels. We found that 46.6% of significant DNAm-to-transcript effects were of positive sign (i.e. DNAm increasing transcription), particularly so when the DNAm site was situated in the gene body (enrichment p-value = $2.15e-10$). This observation is in line with previous genome-wide methylation and gene expression association studies which

reported high fractions of positive correlations (30-41%) \cite{grundberg2013global, wan2015characterization, ruiz2022identification}, and further investigations indicated that the estimated methylation-to-transcript causal effects agree strongly with the respective correlations reported by Grundberg et al. ($P = 2.6e-18$).

14. Four Steiger filtering, the authors mention “trev set at -2” I don’t understand why it is ‘minus’
2. The numerator is a difference of absolute values, where the first should be higher (i.e. relationship with expression) than the second (relationship with outcomes), hence this quantity should be positive? If it is negative, it means that the relationship with the outcome is superior, which is a problem. I am probably missing something, but maybe clarifying the text would be useful.

Indeed, this quantity should be positive and not negative in order to ensure that exposure effects are higher than outcome effects. However, given the measurement errors of individual genetic effect sizes, we only filter out the SNP if the outcome effect is *significantly* larger than the exposure margin. Hence the tolerance margin up to -2, which corresponds to a one sided test p-value threshold of 0.023.

We now clarified that we only remove SNPs for which outcome effects are significantly larger than exposure effects. The corresponding sentence reads as follows:

Prior to the causal effect calculations, IVs were Steiger-filtered to avoid that the IV's effect on Y is significantly larger than it is on E \cite{hemani2017orienting} and were thus required to pass a threshold $t_{rev} < \frac{|\beta_{E_i}| - |\beta_{Y_i}|}{\sqrt{\text{var}(\beta_{E_i}) + \text{var}(\beta_{Y_i})}}$ with t_{rev} set at -2, equivalent to a one sided test p-value threshold of 0.023 \cite{hemani2018mr}.

15. “If the number of mediator-associated instruments was sufficient (≥ 3) to conduct a univariable MR from the mediator on the outcome, we estimated $\alpha_{MY,k}$ from this analysis instead, since computed on a single regressor, narrower CIs are obtained.”
I don’t understand how can the univariate MR can provide the same estimate as the multivariate MR as the effect of mediator on outcome should be adjusted for the exposure in the latter?

As the reviewer correctly points out, univariable MR are not the same as multivariable MR estimates. Thus, when conducting mediation analyses and calculating indirect effects the latter should be used (as demonstrated in our answer to Q6).

However, in case of correlated mediators we may not be able to disentangle individual contribution from each transcript mediator, but still the univariable analysis can provide statistical evidence for the existence of a strong mediator-to-outcome effect, even if we cannot decide which mediator is the driver. Since these reported mediator-to-outcome effects serve to explain biological pathways, we choose to report univariable MR effects whenever possible.

We adapted the phrasing to make this point clearer:

In the estimation of MPs, we were not interested in $\alpha_{MY,k}$ values per se, but we took these effect sizes into account for inferring molecular mechanisms. If the number of mediator-associated instruments was sufficient (≥ 3) to conduct a univariable MR from the mediator to the outcome, we estimated $\alpha_{MY,k}$ from this analysis instead. In fact, the (marginal)

contribution of an individual mediator can be better disentangled in univariable analyses, when mediators are highly correlated.

REVIEWERS' COMMENTS

Reviewer #1 (Remarks to the Author):

Revision

I would like to commend the authors on a thorough revision. I especially like Figure 2. Most of my prior comments have been addressed. I have the following minor comments/questions

1. Although I generally found this clear, I think it would be good to clarify that the finding of an average of 28% of methylation-trait effect mediated by expression does not mean that 28% of total methylation effects on traits are mediated by expression due to not having accounted for differences in total methylation-trait effects or correlation between methylation probes. I think the later interpretation will be tempting for readers so it would be helpful to clarify.
2. Do you have a reference for eq 6 or a justification that can be added to the supplement?
3. The definition of mediation proportion is not restricted to be between 0 and 1 which is a little unintuitive for a quantity referred to as a proportion. It would be nice to add a sentence noting this and describing the interpretation of MP less than 0 or greater than 1.
4. Page 6, the two sentences on lines 78-82 read as contradictory. The first states that including too many mediators did not affect \hat{MP} while the second reads that it did.
5. Page 11 first sentence (line 114) is very confusing.
6. Line 120, is the 3.3 mediators per probe before or after pruning mediators for correlation as described in methods?
7. Figure 5a. I understand from the response how the apparently inconsistent numbers were arrived at. I think other readers may also find this confusing. I would suggest modifying the figure to display the estimated direct effect so that the reader can see how the MP is computed from direct and total effects. I would suggest stating specifically that the displayed estimates of α_{MY} and α_{EM} are inconsistent with the given MP.
8. The estimator in (1) was proposed as the GSMR estimator by Zhu et al (2018) (<https://www.nature.com/articles/s41467-017-02317-2>) so it would be good to cite this.
9. It seems to me that, unlike the Steiger test statistic, t_{rev} is sensitive to trait scaling. Are the betas scaled consistently (e.g. to units of per-sd of E or Y?).
10. I found the description on lines 442 to 461. I would suggest working to clarify this procedure.
11. Line 495, does the regression of $\hat{\theta}_D$ on $\hat{\theta}_T$ include an intercept?
12. Eq. 7 assumes independence between $\hat{\gamma}^{(1)}$ and $\hat{\gamma}^{(2)}$. This probably doesn't hold when comparing different thresholds for P_{EM} .

Reviewer #2 (Remarks to the Author):

The authors have conducted very comprehensive analyses to respond to reviewers comments. I'm satisfied with their responses. I have only few remaining (mostly conceptual) clarifications to ask. The authors now distinguish between (i) overall proportion mediated (MP), when the total effect of DNAm on trait is significant and (ii) detectable MP, when the total effect is significant and the effect of DNAm on the mediator is significant. The two proportions are substantially different, with the detectable MP being higher. I am unsure that the detectable MP is a useful quantity. This is because my understanding of the substantive scientific question is "how much of the effect of DNAm on traits is mediated by transcript levels", which is better assessed by the overall MP. Conversely, the detectable MP answers the question: what is the proportion mediated when we have a strong likelihood that there is a mediator. This I think is much less interpretable and potentially misleading. For example, let's say that instead of going from 2600 probes for (i) to 2000 probes for (ii) we went down to 100

probes. The detectable MP could be very high but the most meaningless as it would concern a very small minority of probes. I therefore suggest that the authors either get rid of the detectable MP or put more emphasis on the overall MP.

In addition, even the overall MP is only 'detectable' in a way as it is estimated only for those probes with a detectable significant total effect. The authors have a technical justification for this choice (i.e. unstable estimation of mediation effects when total effects are null or very small). This is a reasonable justification, but it remains a technical problem which prevents to fully respond to the question, i.e. what is the true MP for all DNAm with a true effect on traits. Clearly, the authors cannot respond to this question as this would require having the true parameters, which we never have. My understanding is that the authors have partially addressed that by varying the threshold of selection of probes with more or less lenient threshold for the total effect, is that correct? I would still appreciate a more nuanced discussion of the limitations of the estimated MPs. In particular, in the limitation section of the manuscript, the authors could highlight again the fact that the overall MP is only estimated when a total effect has been detected and that this might not accurately estimate the true MP. And further remind the reader about the different factors that can lead to either underestimating or overestimating the MPs. Reiterating this in the limitation section is a good practice as interested readers can get a summary of these limitations in one place.

Color code:

Answers to the reviewers are written in green.

Unchanged elements borrowed from the manuscript are in blue (italic), and changed text quoted from the manuscript is in brown (italic).

Reviewer #1

I would like to commend the authors on a thorough revision. I especially like Figure 2. Most of my prior comments have been addressed. I have the following minor comments/questions

We would like to thank the reviewer for the positive assessment of our revisions and appreciate the comments oriented towards clarifying the manuscript.

1. Although I generally found this clear, I think it would be good to clarify that the finding of an average of 28% of methylation-trait effect mediated by expression does not mean that 28% of total methylation effects on traits are mediated by expression due to not having accounted for differences in total methylation-trait effects or correlation between methylation probes. I think the later interpretation will be tempting for readers so it would be helpful to clarify.

We thank the reviewer for raising this point. We now added this limitation to the discussion ("limitation paragraph") which reads as follows:

While our method highlights candidate pathways and provides MP estimates, several limitations are to be considered. First, our MP estimates are based on a selection of 2,623 DNAm-trait pairs with significant total effects ($P_T < 1e-6$), which inherently focuses on DNAm-trait pairs with larger (and hence detectable) effects. In theory, MPs could depend on the magnitude of the total causal effect, thus the reported MP may differ for weaker total effects. A special case of these weaker total effects is when direct and indirect effects differ in sign, leading to a weak total effect with an MP potentially outside the $[0,1]$ range. Furthermore, selected DNAm sites were those with the strongest DNAm-trait signal in their region (up to 1Mb). Thus, we omit secondary methylation signals, which may be mediated by transcript to a different degree. Second, as for all MR-IVW approaches, included IVs might be pleiotropic, ...

2. Do you have a reference for eq 6 or a justification that can be added to the supplement?

We now added a reference (Knuiman et al., 1998) to equation 6 in the method section and the result section where the equation is quoted. This paper explains the origin and derivation of the regression dilution correction term.

3. The definition of mediation proportion is not restricted to be between 0 and 1 which is a little unintuitive for a quantity referred to as a proportion. It would be nice to add a sentence noting this and describing the interpretation of MP less than 0 or greater than 1.

We now clarified the method section "Estimating and comparing mediation proportions", which reads as follows:

Note that \widehat{MP} is an estimator of the true underlying MP and values outside the expected $[0-1]$ range can be observed, especially if $\hat{\theta}_D$ and $\hat{\theta}_T$ estimates are of opposite sign. Such situations

are expected to be rare in our analysis, as the total effect would be expected to be small and hence non-detectable.

4. Page 6, the two sentences on lines 78-82 read as contradictory. The first states that including too many mediators did not affect $\hat{\text{MP}}$ while the second reads that it did.

We thank the reviewer for pointing this out. We are now more explicit about the fact, that we are referring to a downward bias of the MP when irrelevant mediators are included in the model instead of simply stating that there is no bias.

Among a set of 20 potential mediators, those not passing the P_{EM} as determined by univariable MR effects of the exposure on each of these mediators were excluded from the MVMR model (Methods). Using a too lenient or too stringent P_{EM} resulted in downward biased $\hat{\text{MP}}$ s (Fig. 2b), as the former leads to the inclusion of too many non-mediators in the model (giving rise to weak instrument bias), while the latter case fails to include relevant mediators in the model.

5. Page 11 first sentence (line 114) is very confusing.

We rephrased the sentence while also putting a stronger emphasis on our definition of overall mediation.

In addition to the 2,069 DNAm-trait pairs with detectable mediation, there were 554 pairs testable for mediation, but with no detectable causally implicated transcript (Figure 3a). Setting $\hat{\theta}_D$ to $\hat{\theta}_T$ for these pairs and regressing $\hat{\theta}_D$ against $\hat{\theta}_T$ for all 2,623 DNAm-trait pairs combined reduced the $\hat{\text{MP}}$ to 28.3% (95% CI: [26.9%-29.8%]) (Fig. 3d). We refer to this $\hat{\text{MP}}$ as the overall $\hat{\text{MP}}$, as it is a more objective measure of the importance of the transcriptome in mediating DNAm-to-phenotype effects. While more reflective of mediated DNAm effects, it may also be overly conservative since the set of testable transcript mediators ($n = 19,250$ \cite{vosa2021large}) is a magnitude lower than that of the whole transcriptome \cite{howe2021ensembl}.

6. Line 120, is the 3.3 mediators per probe before or after pruning mediators for correlation as described in methods?

3.3 mediators per probe were before pruning mediators for correlation. We clarified this as follows:

The average number of mediator transcripts, potentially correlated, was 3.3 per methylation-trait pair with detectable mediation, indicating that the impact of methylation is not mediated by a single transcript.

Additionally, the sensitivity analysis with uncorrelated mediators (result section: Determining factors of mediation proportions) compares the two numbers (before and after pruning mediators) side-by-side:

The mean number of selected mediators dropped by more than half, from 3.3 to 1.2 (Supplementary Fig. 18), and ...

7. Figure 5a. I understand from the response how the apparently inconsistent numbers were arrived at. I think other readers may also find this confusing. I would suggest modifying the figure to display the estimated direct effect so that the reader can see how the MP is

computed from direct and total effects. I would suggest stating specifically that the displayed estimates of α_{MY} and α_{EM} are inconsistent with the given MP.

We thank the reviewer for pointing this out. We now updated the figure to include the direct effect, and mention in the caption the derivation of MP to avoid confusion.

Figure 5. Plausible DNAm-transcript-trait regulatory mechanisms. **a** Mechanism involving DNAm probe cg10385390, PARK7 and inflammatory bowel disease (IBD). **b** Mechanism involving DNAm probe cg13428477, PDIA5 and platelet count. The top row displays a schematic of the mechanism with calculated univariable (total effect $\hat{\theta}_T$, DNAm-to-transcript effect $\hat{\alpha}_{EM}$ and transcript-to-outcome effect $\hat{\alpha}_{EM}$) and multivariable (direct effect $\hat{\theta}_D$) MR effects (displayed mediation proportions (\hat{MP} s) are derived from $\hat{\theta}_D$ and $\hat{\theta}_T$ estimates). The three following rows show the regional SNP associations ($-\log_{10}(p\text{-values})$) with the trait (GWAS, green), transcript (eQTL, blue) and DNAm (mQTL, brown) probe, respectively. Solid diamonds represent DNAm-associated instruments used in the univariable (for $\hat{\theta}_T$ calculation) and multivariable (for $\hat{\theta}_D$ calculation) MR analyses. Upwards pointing triangles are transcript-associated SNPs that were additionally included in the MVMR instrument set. Red dashed lines indicate the significance thresholds of the respective SNP associations and the vertical black dashed line represents the DNAm probe position. Bottom row illustrates the positions and strand direction of the genes in the locus.

8. The estimator in (1) was proposed as the GSMR estimator by Zhu et al (2018) (<https://www.nature.com/articles/s41467-017-02317-2>) so it would be good to cite this.

We now added the reference to equation (1).

9. It seems to me that, unlike the Steiger test statistic, t_{rev} is sensitive to trait scaling. Are the betas scaled consistently (e.g. to units of per-sd of E or Y?).

All the genetic effect sizes (β_E , β_Y , β_M) were standardized (to SD units) prior to conducting Steiger filtering (i.e., testing against the t_{rev} threshold). Thus, the formula which we explicitly write out in the text corresponds to the Steiger test statistic.

10. I found the description on lines 442 to 461. I would suggest working to clarify this procedure.

To clarify our horizontal pleiotropy simulation analysis, we now support the workflow with a scheme outlining the different steps and visualizing a region with DNAm site and transcript associations. This scheme is introduced and referenced in the corresponding method and result sections and reads as follows:

Method section:

In the following, we outline step-by-step the workflow of the horizontal pleiotropy simulation study for which a schematic representation is shown in Supplementary Fig. 16. First, we considered DNAm-transcript pairs...

1 Multivariable SNP effects on transcript:

$$\beta_{multi} = C_M^{-1} \beta_M \quad C_M: \text{pairwise LD matrix of } m_M \text{ cis-eQTL SNPs}$$

2 For each simulation j (m_M random SNPs \star):

$$\beta_{marginal,j} = C_{E,M,j} \beta_{multi} \quad C_{E,M,j}: \text{LD matrix between } m_E \text{ true and } m_M \text{ random SNPs}$$

"hypothetical transcript effect sizes"

3 MR on hypothetical transcript:

4 Repeat steps 2-3 N_{sim} times

5
$$P_{sim} = \#(P_{EM,j} < P_{EM}) / N_{sim}$$

Supplementary Figure 16. Schematic illustrating the horizontal pleiotropy simulation analysis to assess the possibility of DNAm-to-transcript associations because of horizontal pleiotropy as a result of LD between mQTLs and eQTLs. First DNAm-transcript pairs with a significant MR effect at $P_{EM} < 1e-6$ are selected. Then, multivariable SNP effects on the transcript are calculated based on independent cis-eQTLs (step 1). In each of the following simulations, m_M random SNPs are selected for which marginal SNP-transcript effects are calculated. Note that these hypothetical transcript effect sizes have identical multivariable eQTL effect size distribution as the real transcript (step 2). Next, a univariable MR analysis on this hypothetical transcript yields $P_{EM,j}$ (step 3). Steps 2-3 are repeated N_{sim} times (step 4) which allows to calculate the simulation p-value P_{sim} (step 5).

11. Line 495, does the regression of $\hat{\theta}_D$ on $\hat{\theta}_T$ include an intercept?

The regression does not include an intercept. We now mention this explicitly in the text as follows:

MPs were then calculated by regressing $\hat{\theta}_D$ on $\hat{\theta}_T$ (without intercept) to estimate for the unmediated proportion...

12. Eq. 7 assumes independence between $\hat{\gamma}^{(1)}$ and $\hat{\gamma}^{(2)}$. This probably doesn't hold when comparing different thresholds for P_{EM} .

We agree that equation (7) assumes independence between the gamma estimators (i.e., no correlation between estimators). This may be lenient if there is a positive correlation between two estimators such as when comparing different thresholds for P_{EM} . However, the dependence of these estimators is very complex and estimating their covariance is not straightforward, since we cannot rely on classical bootstrap or jackknife methods. We now add a note to the equation to mention this limitation. This reads as follows:

Of note, this z-test assumes independence between $\hat{\gamma}^{(1)}$ and $\hat{\gamma}^{(2)}$ which is not always guaranteed (i.e., when comparing P_{EM} thresholds), hence the resulting p-values may be lenient.

Reviewer #2 :

The authors have conducted very comprehensive analyses to respond to reviewers comments. I'm satisfied with their responses. I have only few remaining (mostly conceptual) clarifications to ask.

The authors now distinguish between (i) overall proportion mediated (MP), when the total effect of DNAm on trait is significant and (ii) detectable MP, when the total effect is significant and the effect of DNAm on the mediator is significant. The two proportions are substantially different, with the detectable MP being higher. I am unsure that the detectable MP is a useful quantity. This is because my understanding of the substantive scientific question is "how much of the effect of DNAm on traits is mediated by transcript levels", which is better assessed by the overall MP. Conversely, the detectable MP answers the question: what is the proportion mediated when we have a strong likelihood that there is a mediator. This I think is much less interpretable and potentially misleading. For example, let's say that instead of going from 2600 probes for (i) to 2000 probes for (ii) we went down to 100 probes. The detectable MP could be very high but the most meaningless as it would concern a very small minority of probes. I therefore suggest that the authors either get rid of the detectable MP or put more emphasis on the overall MP. In addition, even the overall MP is only 'detectable' in a way as it is estimated only for those probes with a detectable significant total effect. The authors have a technical justification for this choice (i.e. unstable estimation of mediation effects when total effects are null or very small). This is a reasonable justification, but it remains a technical problem which prevents to fully respond to the question, i.e. what is the true MP for all DNAm with a true effect on traits. Clearly, the authors cannot respond to this question as this would require having the true parameters, which we never have. My understanding is that the authors have partially addressed that by varying the threshold of selection of probes with more or less lenient threshold for the total effect, is that correct? I would still appreciate a more nuanced discussion of the limitations of the estimated MPs. In particular, in the limitation section of the manuscript, the authors could highlight again the fact that the overall MP is only estimated when a total effect has been detected and that this might not accurately estimate the true MP. And further remind the reader about the different factors that can lead to either underestimating or overestimating the MPs. Reiterating this in the limitation section is a good practice as interested readers can get a summary of these limitations in one place.

We thank the reviewer for the positive assessment of our revisions and appreciate the comments to better convey our main messages.

We agree with the reviewer that the overall mediation proportion is a more useful quantity to answer the underlying question of what proportion is mediated by transcript levels. After careful examination, we decided to keep both quantities in the manuscript as readers may be interested in both. However, we made adjustments to better point out that only the overall MP answers the biological question of mediated DNAm effects through transcript levels. Changes include the following:

1. Early in the result section ("Overview of the methods"), when we introduce the detectable and overall MP, we emphasize that only the overall MP is reflective of mediated DNAm effects through transcript levels:

We present MP results for DNAm-trait pairs with at least one mediator significantly associated to the exposure ("detectable mediation"), but also for pairs, including the ones without a significant causal effect on any potential transcript ("overall mediation"). The overall MP quantifies more accurately the role of cis-transcripts in mediating DNAm effects, as the

restriction to only DNAm-trait pairs with a mediator could introduce a selection bias towards higher MPs.

2. Furthermore, we reformulated the result section introducing the results of the overall MP to better clarify that these are our main mediation results (result section: “Application to 50 complex traits”). This reads as follows:

In addition to the 2,069 DNAm-trait pairs with detectable mediation, there were 554 pairs testable for mediation, but with no detectable causally implicated transcript (Figure 3a). Setting $\hat{\theta}_D$ to $\hat{\theta}_T$ for these pairs and regressing $\hat{\theta}_D$ against $\hat{\theta}_T$ for all 2,623 DNAm-trait pairs combined reduced the \widehat{MP} to 28.3% (95% CI: [26.9%-29.8%]) (Fig. 3d). We refer to this \widehat{MP} as the overall \widehat{MP} , as it is a more objective measure of the importance of the transcriptome in mediating DNAm-to-phenotype effects. While more reflective of mediated DNAm effects, it may also be overly conservative since the set of testable transcript mediators ($n = 19,250$ \cite{vosa2021large}) is a magnitude lower than that of the whole transcriptome \cite{howe2021ensembl}.

3. In addition to the abstract which only mentions the overall mediation proportion (no change to the previous version), the discussion now also only mentions the overall mediation proportion as follows:

First paragraph:

We presented a framework to quantify mediation of complex trait-impacting effects through an omics layer and demonstrated its application to assayed blood-derived DNAm (exposure) and transcript levels (as mediator). Evidence for mediation of DNAm-to-trait effects through transcripts in cis was found to be at least 28.3% for the 2,623 DNAm-trait pairs with significant total causal effects that could be assessed.

Last paragraph:

Overall, we found solid evidence that almost a third of DNAm-to-complex trait effects are mediated by transcripts in cis.

Regarding our MP estimates, and potential biases introduced by filtering on significant total effects, we now added this as first limitation in the “limitation” section in the discussion. To clarify, we did not perform stratification analyses on the DNAm-trait total effect, and we now mention explicitly in the discussion why we chose to limit our analyses to strong total effects.

This reads as follows:

While our method highlights candidate pathways and provides MP estimates, several limitations are to be considered. First, our MP estimates are based on a selection of 2,623 DNAm-trait pairs with significant total effects ($P_T < 1e-6$), which inherently focuses on DNAm-trait pairs with larger (and hence detectable) effects. In theory, MPs could depend on the magnitude of the total causal effect, thus the reported MP may differ for weaker total effects. A special case of these weaker total effects is when direct and indirect effects differ in sign, leading to a weak total effect with an MP potentially outside the $[0,1]$ range. Furthermore, selected DNAm sites were those with the strongest DNAm-trait signal in their region (up to 1Mb). Thus, we omit secondary methylation signals, which may be mediated by transcript to a different degree. Second, as for all MR-IVW approaches, included IVs might be pleiotropic,

...

As suggested by the reviewer, we made sure to mention all major limitations and factors over- and underestimating MPs in the discussion. Besides the limitation paragraph, we updated the end of the first paragraph of the discussion as follows:

While many robust methods are available for univariable MR, it is not the case for MVMR \cite{carter2021mendelian, sanderson2021multivariable}. Still, we could confirm the robustness of our MVMR estimates through various sensitivity analyses (conditional F-statistic, heterogeneity Q-statistic, excluding the strongest IV) that could not pinpoint any factor drastically biasing our MPs estimates. Importantly, simulation studies indicated that MP estimates were likely to be lower bounds. Low sample size was shown to lead to MP underestimations, as do weak instruments, both for exposure- and mediator-associated IVs.